# No Detail Left Behind:
# Revisiting Self-Retrieval for Fine-Grained Image Captioning

**Manu Gaur** *manugaurwork@gmail.com*
*CVIT, IIIT Hyderabad, India*

**Darshan Singh** *darshan.singh@research.iiit.ac.in*
*CVIT, IIIT Hyderabad, India*

**Makarand Tapaswi** *makarand.tapaswi@iiit.ac.in*
*CVIT, IIIT Hyderabad, India*

**Reviewed on OpenReview:** https://openreview.net/forum?id=gqh0yzPYdo

## Abstract

Image captioning systems are unable to generate fine-grained captions as they are trained on data that is either noisy (alt-text) or generic (human annotations). This is further exacerbated by maximum likelihood training that encourages generation of frequently occurring phrases. Previous works have tried to address this limitation by fine-tuning captioners with a self-retrieval (SR) reward. However, we find that SR fine-tuning has a tendency to reduce caption faithfulness and even hallucinate. In this work, we circumvent this bottleneck by improving the MLE initialization of the captioning system and designing a curriculum for the SR fine-tuning process. To this extent, we present (1) Visual Caption Boosting, a novel framework to instill fine-grainedness in generic image captioning datasets while remaining anchored in human annotations; and (2) BagCurri, a carefully designed training curriculum that more optimally leverages the contrastive nature of the self-retrieval reward. Jointly, they enable the captioner to describe fine-grained aspects in the image while preserving faithfulness to ground-truth captions. Our approach outperforms previous work by +8.9% on SR against 99 random distractors (RD100) (Dessì et al., 2023); and +7.6% on ImageCoDe.

Additionally, existing metrics to evaluate captioning systems fail to reward diversity or evaluate a model's fine-grained understanding ability. Our third contribution addresses this by proposing self-retrieval from the lens of evaluation. We introduce TrueMatch, a benchmark comprising bags of highly similar images that uses SR to assess the captioner's ability to capture subtle visual distinctions. We evaluate and compare several state-of-the-art open-source MLLMs on TrueMatch, and find that our SR approach outperforms them all by a significant margin (*e.g.* +4.8% - 7.1% over Cambrian) while having 1-2 orders of magnitude fewer parameters. We also outperform vanilla SR by +14.4% to +19.5%.

## 1 Introduction

Image captioning, or generating natural language image descriptions, has witnessed remarkable progress over the last decade. Today's captioning systems are composed of sophisticated deep learning architectures (Mokady et al., 2021; Stefanini et al., 2022; Dai et al., 2024; Liu et al., 2023) trained on vast datasets (Lin et al., 2014; Sharma et al., 2018; Thomee et al., 2016; Desai et al., 2021; Schuhmann et al., 2022). However, even with these advances, approaches often generate generic captions that are unable to differentiate similar images (see Figure 1), violating the fundamental purpose of a caption: *to facilitate accurate and efficient communication of visual content* (Fisch et al., 2020; Kreiss et al., 2022; Dessì et al., 2023; Das et al., 2017). We attribute the shortcomings of image captioning systems to three key factors: (1) the nature of their training data, (2) captioning evaluation metrics, and (3) the maximum likelihood estimation (MLE) training approach.

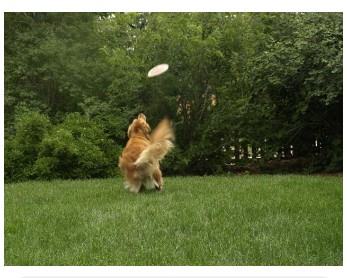
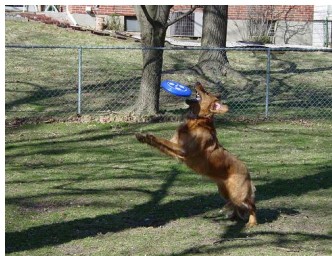
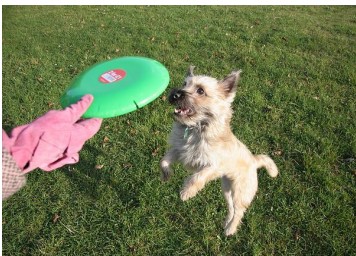

| | | | |
|---|---|---|---|
| **COCO MLE** | A dog jumping in the air to catch a frisbee. | A dog jumping in the air to catch a frisbee. | A dog is holding a frisbee in it's mouth. |
| **COCO SR** *(CVPR 2023)* | A dog jumping in the air to catch a frisbee. | A dog jumping in the air to catch a frisbee. | A dog and a frisbee in a grassy field. |
| **OUR SR** | A brown dog jumps up to catch a **white** frisbee in its mouth as it flies through the air in a grassy field, **surrounded by trees**. | A brown dog jumps up to catch a **blue** Frisbee in its mouth in a grassy field, with a **fence** in the background serving as a boundary for the area. | A small **brown** and **white** dog stands on a grassy field next to a **green** Frisbee, holding it in its mouth and appears to be enjoying playing with it in the grassy field. |

Figure 1: For similar images, captioning systems struggle to generate meaningful captions that uniquely describe each image. In this example, **COCO MLE**: A model trained on COCO with MLE generates the same generic caption for the first two images. **COCO SR** (Dessì et al., 2023): While the self-retrieval (SR) objective may help, the COCO captions are not rich enough to generate salient visual details. **OUR SR**: Our improved data and training recipe results in fine-grained, and therefore discriminant captions.

**I. Challenges with the training data.** Training datasets may be divided into two: curated datasets (COCO (Lin et al., 2014), Flickr30k (Plummer et al., 2015)) or large-scale alt-text data (*e.g.* CC3M (Sharma et al., 2018)). While alt-text is noisy and may be unaligned with the image, human-annotated captions (COCO) may be generic (Kornblith et al., 2023) and lack world knowledge (Bavishi et al., 2023) as annotators describe visual concepts in a simplistic manner (*e.g.* labeling a *Golden Retriever* as a dog). Recently, foundation models are being used to enhance large-scale alt-text data by making them denser (Doveh et al., 2024; Urbanek et al., 2024). However, such methods are prone to inherit biases (*e.g.* gender, geography) present in foundation models (Hall et al., 2023; Sirotkin et al., 2022; Basu et al., 2023; Salman et al., 2022) and exhibit verbose language modeling priors (Liu et al., 2024).

To address these data inadequacies, we propose *Visual Caption Boosting* (VCB), a model agnostic framework designed to generate dense captions that holistically capture different aspects of the image (objects, attributes, relations, scene, *etc.*) while remaining anchored in human annotations (see Section 2). In brief, multiple human annotated captions are blended together using a Large Language Model (LLM) and expanded with an image description generated from a Multimodal Large Language Model (MLLM). In case of conflicting visual details, we prompt the LLM to prefer the blended caption over the description from the MLLM. Thus, VCB creates fine-grained captions that are grounded in human annotations, enabling rich and informative datasets to train image captioning systems.

**II. Image Caption Evaluation**. Current metrics can be broadly classified into reference-based and reference-free. Reference-based metrics such as BLEU (Papineni et al., 2002), CIDEr (Vedantam et al., 2015), and SPICE (Anderson et al., 2016) rely on comparisons to ground-truth (GT) captions that may not include image details beyond salient objects. Although these metrics evaluate grammatical correctness, they tend to penalize captions that are more specific than the ground-truth, often favoring generic descriptions (Wang et al., 2020). Reference-free metrics like CLIPScore (Hessel et al., 2021) alleviate this issue by directly measuring image-text similarity, but may fail to assess discriminativeness of the captions. These limitations necessitate the development of new evaluation strategies that incentivize the models to produce fine-grained captions.

In this work, we ask what makes a "fine-grained" or "good" description when evaluating captioning systems? We posit that a caption should be succinct, but enable a listener to identify the target image from a bag of images with similar visual elements (Liu et al., 2018). To this extent, we propose to evaluate captioning systems through the lens of self-retrieval (SR). We measure the ability to retrieve the target image based on the generated caption within a bag of highly similar distractor images.

Traditionally, improvements in text-to-image (T2I) retrieval have focused on either enhancing the scoring functions (Faghri et al., 2017) or making denser, more detailed text queries (Lai et al., 2023; Li et al., 2024; Singla et al., 2024; Shvetsova et al., 2024). In this work, we introduce a novel perspective: by curating sets of similar images, we can use T2I retrieval as a tool for evaluating whether the text query (caption) captures fine-grained visual details necessary for retrieving the target image. Specifically, we construct *bags* of highly similar images, designed such that uniquely describing each image within a bag demands capturing different facets of visual understanding. For example, retrieving the middle image in Figure 1 requires the caption to incorporate information about attributes (*blue* frisbee) or background objects (*fence*). Note, each caption is generated *without looking at the other images in the bag*. Thus, we use discriminability as a proxy to measure how well a model describes fine-grained components in an image.

While previous works have incorporated SR as a reward signal during training (Dessì et al., 2023), to our knowledge, we are the first to curate bags of similar images and leverage SR to evaluate fine-grained understanding exhibited by captioning systems. To this end, we introduce *TrueMatch*, a benchmark of image sets with varying size. TrueMatch offers a comprehensive evaluation framework to assess the ability of captioning systems to capture various aspects of visual discrimination such as positioning, action, orientation.

**III. Guiding captioners away from their language modeling priors.** MLE training incentivizes captioning systems to overuse common concepts and statistically probable phrases when describing visually similar images. While previous works have optimized self-retrieval (SR) to learn discriminant captioners (Luo et al., 2018; Liu et al., 2018; Cho et al., 2022; Dessì et al., 2023), the models are first trained via MLE on generic caption datasets (*e.g.* COCO). We find that this is suboptimal and show that SR fine-tuning is sensitive to initialization: it is necessary to start with a captioning system that captures fine-grained details in order to better preserve faithfulness to GT captions. In fact, we discover a propensity of captioning systems to hallucinate (Section 4.4) when trained via SR on generic captions (Dessì et al., 2023). Thus, current SR approaches face two distinct challenges: a trade-off between retrieval performance and caption faithfulness, and sub-optimal fine-grained retrieval performance.

To enhance SR's ability to instill fine-grained visual information, we (1) fine-tune both components (language and visual) of the captioning system (Section 4.5), and (2) mine multiple hard negatives (visually similar sets of images) to create training bags. We also introduce a curriculum learning approach that progressively increases the bag size during training to leverage the contrastive nature of our retrieval-based reward (Section 4.6). Our carefully designed curriculum enables SR to leverage the rich initialization provided by *VCB*, improving retrieval performance as well as the faithfulness of the generated captions. Through our training procedure, we are able to circumvent the aforementioned trade-off and achieve substantial performance gains, surpassing (Dessì et al., 2023) by 20% on TrueMatch without making any modifications to the model architecture or reward. Our results demonstrate that a well-crafted training paradigm achieves performance comparable to MLLMs that are orders of magnitude larger. Figure 1 shows images of a bag from TrueMatch and captions generated by the same model architecture trained with different datasets (COCO *vs*. Ours) and paradigms (MLE *vs*. Vanilla SR *vs*. Our SR).

**Key contributions.** We address three challenges plaguing current image captioning systems: *data*, *evaluation*, and *MLE training*. (1) We identify the inadequacies of image captioning datasets in Section 2, and propose *Visual Caption Boosting*, a novel caption enhancement strategy that leverages LLMs and MLLMs to generate dense, informative, and unbiased captions that are anchored in human annotations. (2) We introduce TrueMatch (Section 3), a benchmark consisting of curated *bags* of highly similar images that uses self-retrieval (SR) to assess the ability of captioning systems to capture fine-grained visual distinctions. (3) In Section 4, we leverage the rich data and effective evaluation introduced in the previous two sections to train fine-grained captioning systems with SR. We offer new insights into SR fine-tuning through extensive ablations revealing its sensitivity to MLE initialization, failure to preserve caption faithfulness, and tendency to hallucinate. (4) Finally, we design a simple "plug and play" training recipe that enables SR to improve caption faithfulness while significantly outperforming vanilla SR (Dessì et al., 2023) on TrueMatch and achieving state-of-the-art results on ImageCoDe (Krojer et al., 2022), another benchmark for fine-grained image retrieval.

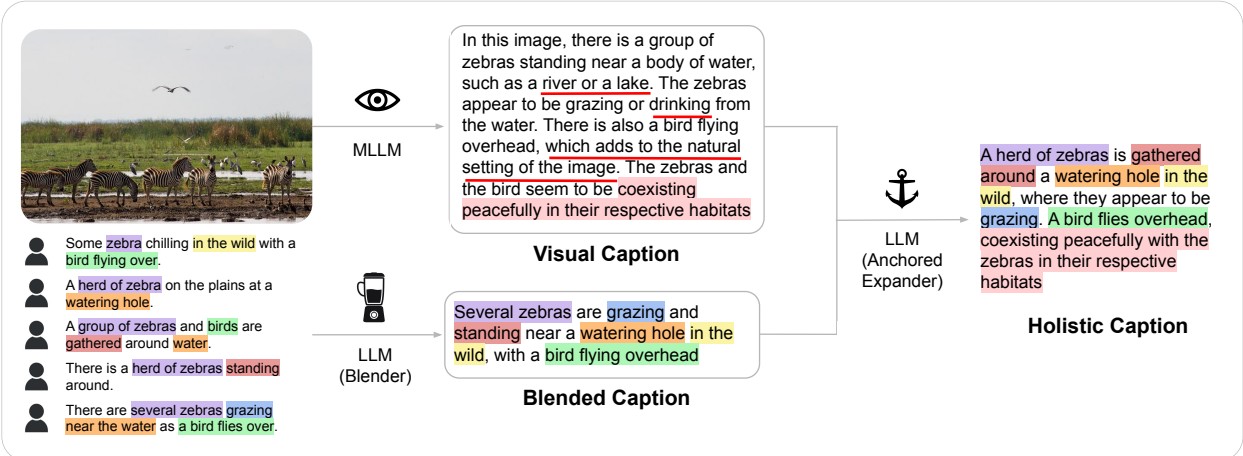

Figure 2: Example of Visual Caption Boosting transforming the original human annotated captions (left) to a Holistic Caption. First, an LLM blends the human annotations to create a Blended Caption. Next, an MLLM generates a dense visual caption that may be noisy. Finally, we create a Holistic Caption by instructing the LLM to incorporate fine-grained details from the Visual Caption with the Blended Caption, while staying anchored in human annotations in case of conflicts. Specific prompts are shared in Appendix A.1. The colors indicate various concepts extracted from the human annotations or the visual caption. The red underlined text (illustrated by us for ease of understanding) indicating hallucinations or verbose text, is ignored in the Holistic Caption as we anchor it to human annotations.

## 2 Improving Datasets through Visual Caption Boosting

Training with dense ground-truth descriptions benefits vision-language models (Doveh et al., 2024; Urbanek et al., 2024; Lai et al., 2023). Although existing image captioning datasets like COCO (Lin et al., 2014) provide multiple annotations per image, they do not elicit detailed captions. This leads to generic annotations such as *"There is a herd of zebras standing around"* (see Figure 2) that ignore finer visual concepts and fail to describe the image holistically. This bottleneck reflects in trained captioning systems as well.

To address this, recent works leverage foundation models to synthetically expand visual information within captions (Doveh et al., 2024; Li et al., 2024; Singla et al., 2024; Lai et al., 2023; Jiao et al., 2024), instilling a wider range of visual details. However these methods are prone to inherit biases present in foundation models (Sirotkin et al., 2022; Basu et al., 2023; Salman et al., 2022). Moreover, the descriptions generated by some of these methods are excessively long making them susceptible to hallucinations (Favero et al., 2024). They also pose challenges to training captioning systems as they exceed the token capacity of current VLMs (*e.g.* CLIP with 77 tokens (Urbanek et al., 2024)). Nevertheless, while individual COCO captions are sparse, we find that they describe complementary facets of the image *e.g. "watering hole", "bird flying over", "herd of zebras"* (Figure 2). This also corroborates findings of Ye et al. (2023) that annotators of different cultural backgrounds describe distinct visual concepts when viewing the same image.

Building upon these findings, we introduce *Visual Caption Boosting* (VCB), a novel two-stage approach to enrich the training data with dense, more informative captions that encourage captioning systems to learn and generate rich descriptions. VCB leverages foundation models and the diverse perspectives offered by human annotators to generate rich descriptions while being anchored in human annotations.

**Step 1. BlendCap** leverages an off-the-shelf LLM to create a blended caption that combines multiple facets of visual information that the human annotators describe. Figure 2 shows an example of how a few captions can be blended together into a comprehensive description of the image using our method. Notably, we prompt the LLM to minimize redundant information resulting in succinct descriptions.

**Step 2. HolisticCap** extends BlendCap by incorporating a fine-grained visual description produced by an MLLM[1]. Specifically, we prompt the LLM to instill the visual caption into the blended caption while preferring human-grounded BlendCap in case of conflicting visual information. As seen in Figure 2, this enables the LLM to ignore hallucinations present in Visual Caption such as *"river or lake"* and *"drinking"*, as they conflict with BlendCap's description like *"watering hole"* and *"grazing and standing"*. Additionally, the anchoring of semantic visual information in human annotations encourages the LLM to eliminate verbose tendencies of MLLMs, producing rich and succinct captions that capture fine-grained details. Although recent works (Urbanek et al., 2024; Lai et al., 2023) address similar challenges, they have some drawbacks as they anchor synthetic visual information in alt-text (Lai et al., 2023) or depend on manual annotation for incorporating dense visual information (Urbanek et al., 2024).

**Anchored captions hallucinate less**. We conduct a human study to measure the degree of hallucinations in BlendCap and HolisticCap (see Appendix A.2). We find that BlendCap introduces negligible hallucinations and that anchoring VisualCap in human annotations significantly (41.6%) reduces hallucinations in HolisticCap.

In summary, Visual Caption Boosting is able to efficiently instill fine-grained visual information into image caption datasets while being less prone to hallucinations. Appendix A.1 presents the LLM and MLLM prompts and Appendix A.3 analyzes the distribution of the lengths of the generated captions.

## 3 TrueMatch: Fine-grained Evaluation through Self-Retrieval

Existing self-retrieval (SR) approaches require models to select the target image from a set of *N random* distractor images in the dataset (Dessì et al., 2023). However, randomly chosen distractors often have simple differences (*e.g.* the primary object or scene), making it easy for captioning systems to distinguish between them. This evaluation is suboptimal as they neither encourage the model to generate detailed captions nor do they evaluate fine-grained abilities of captioning systems.

In this section, we present the SR setup used in our work. We propose *TrueMatch*, a benchmark of carefully curated *bags* of highly similar images that enables SR to evaluate whether captioning systems capture different facets of fine-grained visual discrimination. The results in Section 3.2, show that most captioning systems (including MLLMs) struggle to generate captions that would allow distinguishing fine-grained visual details.

**Self-retrieval setup.** Within a bag of images $\mathcal{B}$ (or a minibatch during training), we require the generated caption $c$ for an image $i$ to retrieve itself (image $i$) from $\mathcal{B}$. Note, $\mathcal{B}$ contains $i$ and a set of visually similar distractor images $\mathcal{D}$. The caption $c$ and all images in $\mathcal{B}$ are encoded using CLIP text and vision encoders respectively and ranking is performed by computing the cosine similarity $\text{sim}(c, i)$ in the CLIP embedding space. Consequently, the caption is deemed to be good $\iff \text{sim}(c, i) > \text{sim}(c, i'), \forall i' \in \mathcal{D}$.

### 3.1 Benchmark Creation

Given a dataset, the benchmark creation process involves curating bags of highly similar images. We use the 10,000 images from COCO's validation and test sets (Karpathy & Fei-Fei, 2015) for our benchmark.

**Creating candidate bags of images.** We use the fine-grained descriptions generated through HolisticCap and encode visual and textual features in the CLIP embedding space. We treat each multimodal embedding (concatenation of the two modalities) as a query and use a simple nearest neighbour search to create bags of highly similar images. Algorithm 1 provides details on the bag creation process, especially for creating bags of variable size. Having bags of varying size facilitates SR evaluation at multiple levels of difficulty as increasing the bag size limits the number visual concepts that can be used to uniquely describe each image. This is different from previous works that typically use image pairs (Jhamtani & Berg-Kirkpatrick, 2018; Park et al., 2019; Tong et al., 2024b) or random distractors (Dessì et al., 2023) to evaluate captioners.

**Automated bag curation.** We compute *intra-bag similarity* $\alpha$, an average pairwise cosine-similarity between multimodal embeddings of all images in the bag to quantify the difficulty of uniquely describing each image in the bag (details in Algorithm 2). To create TrueMatch, we sort the list of bags created

---

[1]We adopted InstructBLIP for this work, see https://huggingface.co/Salesforce/instructblip-vicuna-7b

above in the descending order based on $\alpha$ and use a set $\mathcal{V}$ to keep track of images that are included in TrueMatch. Going down the sorted list, a bag is added to the benchmark only if none of its images exist in $\mathcal{V}$. This ensures that only the hardest bags with highest visual similarity are added to the benchmark for each visual concept in the embedding space. Bags within TrueMatch unlike previous works involving image sets, are fine-grained (Dunlap et al., 2024) and capture different aspects of visual discrimination beyond object positioning and negation (Krojer et al., 2022; Jhamtani & Berg-Kirkpatrick, 2018; Park et al., 2019).

**A manual filtering** pass is performed to retain only those bags that would require captioning systems to capture some aspect of fine-grained visual discrimination. For bag size 3, this reduces the number of bags in TrueMatch from 680 to 254. Some examples of selected and discarded bags are shown in Figure 10.

## 3.2 Experiment 1: Results on TrueMatch

We evaluate several open-source captioning approaches, MLLMs, and SR trained models on TrueMatch. Recall@1 (R@1) is reported in Table 1 for bag sizes 3, 5, and 7.

**Approaches lack fine-grained details.** Captioning models, irrespective of their size, struggle to capture fine-grained visual details leading to poor performance on TrueMatch.

**Doing more with less**. Although billion-parameter MLLMs have achieved impressive results (Liu et al., 2023; Tong et al., 2024a), Table 1 demonstrates that they still struggle with fine-grained visual discrimination. In fact DiscriTune (Dessì et al., 2023), that trains ClipCap with the vanilla SR setup matches InstructBLIP despite being two orders of magnitude smaller. Cambrian-1 is the best-performing open-source model and although it surpasses DiscriTune,

Table 1: Recall@1 for several open-source captioning models, MLLMs, and SR-based methods on TrueMatch. The number of bags in #3 is 254, #5 is 104, and #7 is 93.

| Method | Params | TrueMatch | | |
|---|---|---|---|---|
| | | #3 | #5 | #7 |
| Random chance | - | 33.3 | 20.0 | 14.3 |
| OFA (Wang et al., 2022) | 180M | 50.4 | 36.3 | 37.2 |
| ClipCap (Mokady et al., 2021) | 240M | 50.8 | 36.5 | 33.8 |
| CoCa (Yu et al., 2022) | 640M | 52.2 | 40.4 | 38.2 |
| PaliGemma-224 (Beyer et al., 2022) | 3B | 48.3 | 35.6 | 32.7 |
| PaliGemma-448 (Beyer et al., 2022) | 3B | 49.3 | 38.5 | 33.3 |
| InstructBLIP (Dai et al., 2024) | 13B | 53.7 | 42.7 | 42.1 |
| LLaVA 1.6 (Liu et al., 2023) | 34B | 57.9 | 46.9 | 47.9 |
| Cambrian-1 (Tong et al., 2024a) | 3B | 58.3 | 48.8 | 52.8 |
| Cambrian-1 (Tong et al., 2024a) | 8B | **60.6** | **53.3** | **53.1** |
| Discritune (Dessì et al., 2023) | 240M | 53.3 | 42.3 | 38.4 |
| Ours (best) | 240M | **67.7** | **58.7** | **57.9** |

our proposed approach outperforms it by a significant margin. This demonstrates the effectiveness of both: TrueMatch for evaluating captioning systems, and our approach of using SR to improve captioning (Section 4).

## 3.3 Experiment 2: Benchmarking Visual Caption Boosting with Self-Retrieval

We evaluate the quality of captions adopted in VCB in Table 2. Along with TrueMatch, we also adopt the SR evaluation strategy of Dessì et al. (2023) with 99 random distractors (RD100). On RD100, BlendCap outperforms original COCO captions by a large margin of 8% on R@1 and 6.2% on ClipScore, confirming that human annotations capture complementary visual aspects of the same image. However, this sizeable gap shrinks to 2.5% on TrueMatch (bags of size #3). This confirms that even though annotated captions blended together work better, they inherently lack fine-grained details (see Figure 2) necessary to perform well on TrueMatch. HolisticCap, on the other hand, yields remarkable performance gains over

Table 2: R@1 scores for COCO and VCB captions evaluated on RD100 (100 random distractors) and TrueMatch.

| Caption | RD100 | | TrueMatch | | |
|---|---|---|---|---|---|
| | R@1 | ClipSc | #3 | #5 | #7 |
| COCO | 80.9 | 26.4 | 51.1 | 41.3 | 39.6 |
| VisualCap | 86.9 | 32.2 | 53.7 | 42.7 | 42.1 |
| BlendCap | 88.9 | 32.6 | 53.6 | 44.8 | 43.4 |
| HolisticCap | **91.3** | **33.4** | **57.5** | **48.1** | **49.1** |

COCO and BlendCap across both setups: RD100 and all bag sizes of TrueMatch. This demonstrates the effectiveness of VCB in instilling fine-grained information into standard image captioning datasets.

We study the effectiveness of anchoring visual captions in producing more succinct (see Appendix A.3) and fine-grained captions. Table 2 shows that standalone VisualCap (generated using the MLLM) is slightly worse than BlendCap. However, anchoring VisualCap in human annotations to create HolisticCap results

in +4% improvement over BlendCap on TrueMatch. Additionally, by drawing comparisons to Table 1, we observe that HolisticCap achieves performance on par with larger (LLaVa 1.6) and newer (Cambrian-1-3B) MLLMs compared to InstructBLIP.

> ***Finding* 1.** Self-retrieval with *TrueMatch* evaluates models' ability to produce fine-grained captions and *Visual Caption Boosting* is a promising way to enrich captioning datasets with granular details.

In summary, SR within TrueMatch enables fine-grained evaluation of captioning systems, probing different facets of visual discrimination. It requires captions to be unique even for semantically similar images, thereby addressing the limitations of traditional captioning metrics.

## 4   Improving Captioning Systems with Self-Retrieval as a Training Objective

In this section, we present key insights on improving training with self-retrieval (SR), resulting in significant performance gains over previous works (Dessì et al., 2023). We begin by outlining the method (Section 4.1) and experimental setup (Section 4.2). Our experiments reveal that captioners trained with SR are highly sensitive to their MLE initialization (Section 4.3). In fact, we discover a trade-off between caption faithfulness and retrieval performance plaguing captioning systems fine-tuned with previous SR approaches (Dessì et al., 2023; Liu et al., 2018). To address this we: (1) initialize our model with more detailed captions from HolisticCap, (2) fine-tune the visual encoder with SR (Section 4.5), and (3) mine hard training bags and design a curriculum over bag sizes (Section 4.6). Finally we show that our training strategy is complementary to CIDEr optimization (Rennie et al., 2017) resulting in further improvements (Section 4.8).

### 4.1   Methodology

**Model architecture.** Similar to Dessì et al. (2023), we adopt ClipCap (Mokady et al., 2021), a lightweight simplification of modern MLLMs (*e.g.* LLaVA, InstructBLIP, Cambrian). ClipCap connects a pretrained visual encoder (CLIP (Radford et al., 2021)) to a pretrained language model (GPT-2 (Radford et al., 2019)) through a simple MLP adapter. The adapter is tasked with mapping the rich visual embeddings from CLIP into a fixed number of *prefix tokens*. These tokens capture essential visual information and guide the language model towards generating an image-conditioned caption.

Training this captioning system has two steps: (1) model pretraining with maximum likelihood estimation (MLE), and (2) model fine-tuning by maximizing the SR reward with REINFORCE (Williams, 1992). We outline both training objectives, followed by an experimental investigation of their properties and limitations.

**Maximum Likelihood pretraining** models the training data distribution. Specifically, captioning models are often teacher-forced to learn the word (token) distribution that maximizes the log-likelihood of the ground-truth captions given an input image. However, this results in strong language modeling priors resulting in generic captions that use a small vocabulary (see Section 4.7).

**Maximizing self-retrieval with Reinforce.** A fine-tuning step that optimizes a metric (*e.g.* CIDEr) is popular in training captioners (Rennie et al., 2017). We adopt Dessì et al. (2023)'s contrastive formulation of SR that compares the generated caption against distractors, providing an optimal learning signal:

$$\text{Reward } R(c, i, \mathcal{D}) = \log \frac{e^{\text{sim}(c,i)}}{\sum_{i' \in \mathcal{D} \cup \{i\}} e^{\text{sim}(c,i')}}. \tag{1}$$

We use REINFORCE to optimize the captioning system by estimating the gradient of the expected reward $\mathbb{E}_{c \sim P} \left[ \mathcal{R}(c, i, D) \nabla_\theta \log P(c|i; \theta) \right]$, since backpropagation is broken due to auto-regressive sampling. Further discussion on using REINFORCE and variance reduction of the gradient estimates is in Appendices C.1 and C.2.

### 4.2   Experiment Setup and Metrics

We present results for all our experiments on ClipCap. During MLE pretraining, the ClipCap MLP adapter is trained from scratch, CLIP is frozen, and GPT-2 is fine-tuned. When fine-tuning with REINFORCE, ClipCap's

MLP adapter is trained normally while CLIP and GPT-2 are fine-tuned with LoRA adapters (Hu et al., 2021) with rank 32. The model is trained with the AdamW optimizer (Loshchilov & Hutter, 2018) on a single A6000 GPU. Further details about the hyperparameters are provided in Appendix C.3.

We evaluate the quality of captions with NLG metrics: BLEU-4 and CIDEr, keeping the reference set consistent with the training distribution. On RD100, a setup with 99 random distractors (Dessì et al., 2023), we use R@1 and ClipScore. On TrueMatch, we report R@1.

### 4.3 Experiment 3: Improving SR with Richer MLE Initialization

Following the findings of Section 3.3, we use TrueMatch to investigate the impact of different MLE initializations on SR fine-tuning in Table 3. We study SR in the same setting as previous work (Dessì et al., 2023): (1) we only fine-tune the language model (SR-L), but using LoRA adapters; and (2) until Section 4.6 we use 99 random distractors to fine-tune with the SR reward.

**MLE pretraining yields generic captions.** Table 3 rows 1-3 report the performance of MLE training on captioning datasets with increasing level of detail. We observe that improving the training captions results in a significant performance boost. Training with HolisticCap beats COCO by +8.2% on RD100 R@1 and +5.1% on TrueMatch #3. However, comparing performance against the ground-truth HolisticCap directly (from Table 2), we see

Table 3: Results for different training methods and captions show the effectiveness of improved initialization with HolisticCap and SR-L.

| | MLE Dataset | Training | CIDEr | RD100 | | TrueMatch | | |
|---|---|---|---|---|---|---|---|---|
| | | | | R@1 | ClipSc | #3 | #5 | #7 |
| 1 | COCO | | 107.8 | 76.5 | 29.9 | 50.8 | 36.5 | 33.8 |
| 2 | BlendCap | MLE | 80.2 | 84.9 | 31.6 | 51.8 | 42.5 | 42.2 |
| 3 | HolisticCap | | 26.6 | 84.7 | 31.7 | 55.9 | 43.7 | 42.2 |
| 4 | COCO | | 108.2 | 83.7 | 30.2 | 53.3 | 42.3 | 38.4 |
| 5 | BlendCap | SR-L | 80.1 | 87.5 | 31.7 | 54.7 | 47.5 | 44.2 |
| 6 | HolisticCap | | 27.8 | **87.6** | **31.9** | **59.1** | **47.7** | **46.5** |

that MLE training on HolisticCap yields -6.6% on RD100 R@1 and -1.6% on TrueMatch #3. This indicates that even though HolisticCap improves performance, MLE training (i) generates generic descriptions that may not be aligned with the image content, and (ii) leads to object hallucinations (see Appendix C.7).

**Self-retrieval fine-tuning benefits from VCB.** SR-L reduces MLE's preference towards statistically probable phrases resulting in more discriminant captions, reduced object hallucinations and consistent improvements in RD100 and TrueMatch across datasets (compare rows 1-4, 2-5, 3-6). Providing a richer initialization to SR-L through VCB enhances fine-grained visual discrimination. Row 6 outperforms DiscriTune (row 4) (Dessì et al., 2023) with +3.9% RD100 and +5.8% TrueMatch #3. This illustrates the importance of a good MLE initialization for SR and the effectiveness of *VCB*.

**Longer reference sets hurt CIDEr.** We see a significant drop in CIDEr scores for BlendCap and HolisticCap compared to COCO in Table 3. Upon further analysis, we believe that CIDEr rewards brevity. Appendix C.6 shows some valid short captions that unfortunately obtain scores close to 0.

**Self-retrieval unlocks latent semantic information when fine-tuning the LLM.** Although BlendCap does not capture any new information that is not present in one of the COCO annotations, being descriptive is sufficient to achieve superior retrieval performance against random distractors (BlendCap improves over COCO by +8% on RD100 R@1, Table 2). Further, MLE pretraining is able to model these data distributions as the performance gap is maintained at +8.4% on RD100 R@1 between rows 1 and 2 in Table 3. The model trained on COCO, due to the MLE objective, generates sparse captions that resemble the independent annotations of COCO. Thus, even though the semantic information in the embedding spaces of both models (trained on COCO or BlendCap) are similar, their ability to access this information is bottlenecked by the MLE objective, leading to a large gap between COCO and BlendCap on RD100 in Table 3.

Interestingly, SR-L fine-tuning narrows this gap dramatically from +8.4% to a mere +3.8% (rows 4, 5). This indicates that SR fine-tuning teaches the captioner to uncover semantic information within its embedding space, even when it was initially obscured by MLE. Furthermore, the mostly unchanged CIDEr scores hint that the captioner remains faithful to the GT captions for both COCO (rows 1, 4) and BlendCap (rows 2, 5).

This is remarkable, as it demonstrates that SR fine-tuning unlocks latent semantic information and steers the language model while preserving the initial data distribution.

> **_Finding_ 2.** Self-retrieval steers the language model to generate discriminative captions by uncovering latent semantic information that was initially obscured by MLE.

### 4.4 Trade-off plaguing SR fine-tuning: Retrieval Performance *vs*. Caption Faithfulness

While SR fine-tuning removes language modeling priors from captioners by making them more discriminant, we observe that they have a tendency to become less faithful to the GT captions upon extended training. We contrast these two tendencies of SR fine-tuning in Figure 3, comparing CIDEr and RD100 R@1 scores. We train for 100 epochs compared to the default 20 epochs used for all other experiments.

While SR performance continually improves (+7.9%) with extended training, a corresponding degradation in CIDEr (-15.4%) is observed. This trade-off reveals that SR has the capacity to elicit better retrieval from the captioning system, even at the cost of generating *lower-quality captions*. We further investigate with qualitative examples in Appendix C.8. Dessì et al. (2023) attribute this deterioration to the model diverging from the GT distribution during SR-L fine-tuning and argue that it is desirable. While this is partially true (see Figure 7), our investigation also reveals that captioners trained with vanilla SR-L, in a dash to enhance retrieval performance, build a propensity to hallucinate attributes (see Figure 8) resulting in poor CIDEr scores. This tendency is only exacerbated with extended training resulting in further degradation of NLG metrics and increased object hallucinations (see Appendix C.7). These findings reveal an important bottleneck in SR fine-tuning:

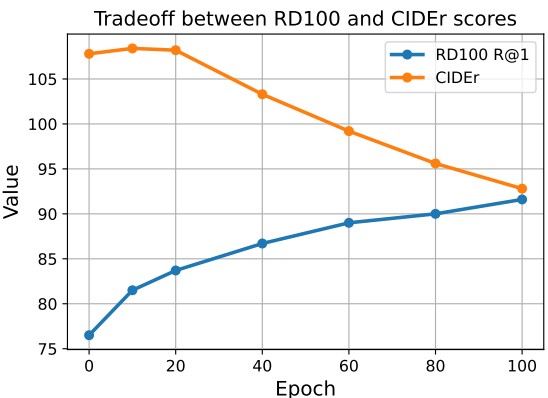

Figure 3: RD100 R@1 continually increases while CIDEr degrades when fine-tuning ClipCap with SR-L on COCO for 100 epochs.

> **_Finding_ 3.** Captioners fine-tuned with SR-L suffer from a trade-off between retrieval performance and faithfulness, often hallucinating details and deviating from GT captions with extended training.

Finally, the continued improvement in RD100 R@1 over 100 epochs indicates that vanilla SR-L fine-tuning fails to saturate SR's capacity to instill fine-grained visual discrimination in captioning systems. This motivates us to explore better SR fine-tuning strategies while mitigating their propensity to hallucinate. Since a higher learning rate destabilizes the policy gradients, we turn to: (1) fine-tuning the visual encoder (CLIP) with SR; and (2) adopting a curriculum over bags of hard negatives.

### 4.5 Experiment 4: Fine-tuning CLIP with Self-Retrieval

We present a comprehensive analysis of fine-tuning language and vision modules of the captioning system with LoRA in Table 4. Fine-tuning CLIP (SR-V) makes the captions discriminative. For example, with BlendCap MLE pretraining, SR-V fine-tuning results in +2.6% on RD100 R@1 and +6.6% on TrueMatch #3 (row 4, 5). However this comes at the cost of deteriorating NLG metrics (-7.7% CIDEr), suggesting that SR-V is unable to preserve the faithfulness of generated captions. We further verify that even after extended SR-L fine-tuning to match SR-V's retrieval scores, SR-L has a higher CIDEr (see Appendix C.4). This suggests that CIDEr deteriorates due to CLIP fine-tuning and not due to NLG metrics penalizing discriminative captions (Wang et al., 2020). This is a notable bottleneck in fine-tuning CLIP with SR: while it enables superior retrieval performance, it makes the captioner less faithful to the GT captions.

Further qualitative analysis of the captions generated by SR-L and SR-V in Figure 9, reveal that captioners fine-tuned with SR-V struggle to bind attributes to the correct objects. For instance, while SR-V captures the *red* color of the *forklift*, it binds it with the incorrect object (*train*). This is likely due to two reasons: (1) the retrieval objective treats the caption as a bag of words and does not promote compositionality; and (2) CLIP has been shown to struggle with attribute binding (Lewis et al., 2022; Hsieh et al., 2024).

**RD100 is suboptimal for training fine-grained captioners.** On TrueMatch, while SR-V fine-tuning provides substantial improvements over SR-L for COCO and BlendCap (rows 1-2, 4-5), we observe relatively smaller gains for HolisticCap (rows 7-8). This indicates that the vanilla SR-L objective of retrieving against 99 random distractors (RD100) is not challenging enough to provide a good learning signal for models pretrained on fine-grained HolisticCap. While SR-LV does better than SR-V on HolisticCap (rows 8-9), the gap between BlendCap

Table 4: Ablation for fine-tuning different modules with SR: GPT-2 (SR-L), CLIP (SR-V), both (SR-LV). Row 1 is Dessì et al. (2023). **Bold** indicates best, *italics* is second. (x) presents improvement against SR-L.

| | MLE Dataset | Training | NLG C | RD100 R@1 | TrueMatch #3 | TrueMatch #5 | TrueMatch #7 |
|---|---|---|---|---|---|---|---|
| 1 | COCO | SR-L | 108.2 | 83.7 | 53.3 | 42.3 | 38.4 |
| 2 | COCO | SR-V | 97.2 | 86.1 (2.4) | 59.3 (6.0) | 49.2 (6.9) | 45.0 (6.6) |
| 3 | COCO | SR-LV | 96.6 | 87.4 (3.7) | 59.6 (6.3) | 51.2 (8.9) | 47.2 (8.8) |
| 4 | BlendCap | SR-L | 80.1 | 87.5 | 54.7 | 47.5 | 44.2 |
| 5 | BlendCap | SR-V | 72.4 | 90.1 (2.6) | 61.3 (6.6) | **52.9** (5.4) | 51.0 (6.8) |
| 6 | BlendCap | SR-LV | 72.7 | **90.7** (3.2) | 61.3 (6.6) | 52.3 (4.8) | 50.1 (5.9) |
| 7 | HolisticCap | SR-L | 27.8 | 87.6 | 59.1 | 47.7 | 46.5 |
| 8 | HolisticCap | SR-V | 25.6 | *90.2* (2.6) | *61.9* (2.8) | 52.5 (4.8) | **51.5** (5.0) |
| 9 | HolisticCap | SR-LV | 26.5 | **90.7** (3.1) | **62.6** (3.5) | *52.7* (5.0) | *51.2* (4.7) |

(row 6) and HolisticCap (row 9) still remains small. To address this, we devise a curriculum with hard negatives, discussed in the next section, to increase the difficulty of SR fine-tuning objective.

> *Finding* **4.** Fine-tuning the visual module with self-retrieval instills fine-grained visual details in the captioning system, while increasing hallucinations and struggling with attribute binding.

Among methods in Table 4, SR-LV demonstrates the best retrieval performance while also preserving CIDEr slightly better than SR-V. Thus, we adopt SR-LV for subsequent experiments with a training curriculum.

### 4.6 Experiment 5: Self-Retrieval Fine-tuning with Bags and Curriculum

The above SR experiments were trained using 99 random distractors (Dessì et al., 2023). Instead, in this experiment, we fine-tune with bags of highly similar images within a training minibatch (see Appendix B.2). Retrieving against multiple hard negatives flattens the softmax distribution resulting in a stronger learning signal. We also propose a curriculum over bag sizes (BagCurri) to more optimally leverage the contrastive reward (see Appendix B.3). SR fine-tuning with BagCurri instills fine-grained visual details in the model, essential for discriminating highly similar images.

Table 5: Ablation of fine-tuning with the bag curriculum (BagCurri) as compared against vanilla SR (SR-L). Row 1 is Dessì et al. (2023).

| | MLE Dataset | Training | Bag Curri | NLG CIDEr | RD100 R@1 | TrueMatch #3 | TrueMatch #5 | TrueMatch #7 |
|---|---|---|---|---|---|---|---|---|
| 1 | COCO | SR-L | - | 108.2 | 83.7 | 53.3 | 42.3 | 38.4 |
| 2 | COCO | SR-L | ✓ | 103.4 | 84.5 | 53.7 | 46.2 | 41.5 |
| 3 | COCO | SR-V | ✓ | 85.8 | 87.4 | 63.1 | 51.9 | 49.3 |
| 4 | COCO | SR-LV | ✓ | 84.2 | 85.8 | 67.2 | 56.3 | 53.6 |
| 5 | Holistic-Cap | SR-L | - | 27.8 | 87.6 | 59.1 | 47.7 | 46.5 |
| 6 | Holistic-Cap | SR-L | ✓ | 31.3 | 88.5 | 58.4 | 50.0 | 49.5 |
| 7 | Holistic-Cap | SR-V | ✓ | 25.8 | 90.6 | 64.0 | 54.6 | 54.1 |
| 8 | Holistic-Cap | SR-LV | ✓ | 29.7 | 91.3 | 65.2 | 57.1 | 57.1 |

**SR-V versus SR-L with our bag curriculum.** We evaluate the impact of fine-tuning different components of the captioning system (CLIP and GPT-2) with BagCurri compared to vanilla SR-L in Table 5. We find that SR-V fine-tuning with our curriculum forces CLIP to learn fine-grained visual features resulting in a substantial +5% to +11% improvement over vanilla SR-L (rows 3 *vs.* 1, 7 *vs.* 5) on TrueMatch. Interestingly, SR-V with BagCurri renders COCO (row 3) comparable to HolisticCap (row 7) on TrueMatch #3. However, these disproportionate gains for COCO come at the expense of CIDEr

scores, which plummet -22.4% compared to SR-L. Further qualitative analysis of the generated captions shows that SR-V fine-tuning with BagCurri amplifies the trends observed in Section 4.5 – while it boosts retrieval performance, it exacerbates the inability of the captioner to bind attributes to the correct objects and increases hallucinations (see Figure 9). In contrast, BagCurri may be too challenging for SR-L fine-tuning, as evident by the small improvements of +0% to +4% over vanilla SR-L (rows 2 *vs.* 1, 6 *vs.* 5) on RD100 or TrueMatch. Nevertheless, similar to the findings of Section 4.3, we find that fine-tuning SR-L with BagCurri benefits heavily from the rich initialization provided by VCB. While the CIDEr score drops for COCO when using SR-L with BagCurri (-4.8%, row 2 *vs.* 1), for HolisticCap we observe an improvement (+3.5%, row 6 *vs.* 5).

**Best of both worlds: SR-LV with BagCurri.** The above findings indicate that BagCurri has the capacity to enhance both: (1) visual discrimination ability of the captioner with SR-V; and (2) adherence to GT captions with SR-L when provided with a rich initialization. Motivated by this, we initialize the captioner with HolisticCap and fine-tune both the language and visual components (SR-LV) with BagCurri (row 8), retaining the best of both worlds. SR-LV+BagCurri further enhances SR-V's already impressive performance on TrueMatch (achieving +6-10% over vanilla SR-L) while also improving the CIDEr score over vanilla SR-L by +1.9% (rows 5, 8). This is momentous, as without modifying the model's architecture or reward function, we outperform vanilla SR-L (Dessì et al., 2023) by an average +15% across all bag sizes on TrueMatch (row 1, 8), while circumventing the caption faithfulness trade-off plaguing current SR approaches.

**Importance of the curriculum.** In Appendix B.3, we investigate the role our curriculum plays in overcoming the retrieval and caption quality trade-off compared to directly using bags as hard negatives. Table 10 reveals that BagCurri is responsible for preserving caption faithfulness, as indicated by a decrease in CIDEr when bags are used without a curriculum.

> ***Finding* 5.** Providing a rich initialization with VCB and SR-LV fine-tuning with BagCurri instills fine-grained visual details in the captioner while preserving its faithfulness to GT captions.

### 4.7 Generating Diverse Captions

We evaluate how VCB and SR increase caption diversity by measuring the number of words that appear $\geq 5$ times on the COCO test set. In Table 6, we observe a clear increase in the number of unique words as we progress from COCO to BlendCap to HolisticCap, across all optimization and training strategies. Furthermore, within each individual dataset, there is a substantial increase in the usage of diverse vocabulary with transi-

Table 6: Number of words with frequency $>= 5$ on the COCO test set for captioners trained with different optimization strategies, modules, and datasets.

| Dataset | MLE | Self-Retrieval | | | +BagCurri |
| | | SR-L | SR-V | SR-LV | SR-LV |
|---|---|---|---|---|---|
| COCO | 428 | 689 | 719 | 762 | 770 |
| BlendCap | 1220 | 1306 | 1345 | 1366 | 1435 |
| HolisticCap | 1431 | 1535 | 1606 | 1628 | 1732 |

tions from MLE to SR-L, SR-V, and SR-LV. Notably, SR-LV with BagCurri yields significant improvements over MLE: relative +80% on COCO, +17.6% on BlendCap, and +21% on HolisticCap. This demonstrates the effectiveness of VCB and our curriculum in guiding the captioning system away from the language modelling priors and improving caption diversity.

### 4.8 Experiment 6: Combining Self-Retrieval and CIDEr Optimization

CIDEr optimization with REINFORCE is commonly used to improve captioning system's adherence to the GT captions (Rennie et al., 2017). As the final experiment, we present results of combining it with SR in Table 7.

**CIDEr optimization** (rows 3, 8) improves faithfulness to GT captions compared to MLE pretraining (rows 1, 6): CIDEr improves +10.7% on COCO and +29.1% on HolisticCap. Importantly, CIDEr optimization on top of HolisticCap MLE pretraining provides +4.8% on TrueMatch #7 (rows 6, 8), while we see only a smaller +2.5% with COCO captions. Even without SR fine-tuning, the larger improvement with HolisticCap emphasizes the importance of MLE initialization with captions containing fine-grained visual details (VCB). Furthermore, compared to vanilla SR-L (rows 2, 7), CIDEr opt. (rows 3, 8) fairs worse on both RD100 and TrueMatch. This presents an opportunity to optimize both rewards simultaneously (Luo et al., 2018).

**CIDEr meets SR.** Building upon the findings of previous sections, we push the boundaries of SR by explicitly adding CIDEr to our discriminative reward $R = SR + \lambda \cdot CIDEr$. We conduct an ablation over various $\lambda$ values (see Appendix C.5), and select $\lambda = 0.5$ for our experiments.

With HolisticCap, we see that joint optimization (row 10) outperforms the most discriminant SR model (row 9) achieving improvements on all metrics: +8.3% on CIDEr, +1.3% on RD100 R@1, and +0.8% to +2.5% on TrueMatch. Conversely, on COCO captions (rows 4, 5), we observe that adding CIDEr optimization adversely affects SR metrics on TrueMatch with -5.1% to -7.6% performance drops. This aligns with previous findings and underscores the importance of initialization during SR fine-tuning.

Table 7: Impact of combining SR with CIDEr optimization. Results are presented for various combinations of RL rewards. C: CIDEr, SR: Self-Retrieval, and BC: SR with BagCurri. Row 1 is Mokady et al. (2021), row 2 is Dessì et al. (2023), and row 3 is Rennie et al. (2017) with ClipCap.

| | MLE Dataset | C | Reward SR | BC | NLG CIDEr | RD100 R@1 | TrueMatch #3 | #5 | #7 |
|---|---|---|---|---|---|---|---|---|---|
| 1 | | - | - | - | 107.8 | 76.5 | 50.8 | 36.5 | 33.8 |
| 2 | | - | SR-L | - | 108.2 | 83.7 | 53.3 | 42.3 | 38.4 |
| 3 | COCO | ✓ | - | - | 118.5 | 79.1 | 50.7 | 37.3 | 36.3 |
| 4 | | - | SR-LV | ✓ | 84.2 | 85.8 | *67.2* | 56.3 | 53.6 |
| 5 | | ✓ | SR-LV | ✓ | 92.8 | 89.0 | 59.6 | 51.2 | 47.2 |
| 6 | | - | - | - | 26.6 | 84.7 | 55.9 | 43.7 | 42.2 |
| 7 | Holistic- | - | SR-L | - | 27.8 | 87.6 | 59.1 | 47.7 | 46.5 |
| 8 | Cap | ✓ | - | - | **55.7** | 86.9 | 57.6 | 47.1 | 47.0 |
| 9 | | - | SR-LV | ✓ | 29.7 | *91.3* | 65.2 | *57.1* | *57.1* |
| 10 | | ✓ | SR-LV | ✓ | *38.0* | **92.6** | **67.7** | **58.7** | **57.9** |

Table 7 row 10 is our best model that also achieves highest performance on TrueMatch as compared to several MLLMs that are orders of magnitude larger and trained on more data (see Table 1).

## 5 Generalization Experiments

In this section, we evaluate the generalization of our approach on Image-CoDe (Krojer et al., 2022), a benchmark for text-based image retrieval where image sets are created with 10 sequential video frames. Different from TrueMatch, ImageCoDe focuses on fine-grainedness found across video frames such as negation, occlusion, *etc*. We follow the same experimental setup adopted by Dessì et al. (2023). Table 8 shows that SR fine-tuning of both the vision and text modules with our bag curriculum (SR-LV BagCurri) achieves state-of-the-art results. In fact, it outperforms Dessì et al. (2023)'s work on using SR-L after MLE pretraining on COCO (row 2, +7.6%) or Conceptual Captions, a 30× larger dataset (row 3, +1.7%).

Table 8: Zero-shot evaluation on the ImageCoDe validation set (R@1). Rows 2 and 3 are from Dessì et al. (2023).

| | Dataset | Method | R@1 |
|---|---|---|---|
| 1 | COCO | MLE | 28.8 |
| 2 | COCO | SR-L | 30.3 |
| 3 | ConCap | SR-L | 36.2 |
| 4 | | MLE | 31.9 |
| 5 | HolisticCap | SR-L | 33.0 |
| 6 | | SR-LV BagCurri | **37.9** |

## 6 Conclusion

We investigated self-retrieval (SR) as a way to comprehensively improve current captioning systems on all three fronts: (1) data, (2) evaluation, and (3) training. (1) We proposed Visual Caption Boosting (VCB), a model agnostic framework to instill fine-grained visual information into generic image captioning datasets while remaining anchored in human annotations. (2) Furthermore, we created TrueMatch, a benchmark comprised of highly similar image bags. SR evaluation with TrueMatch enabled fine-grained evaluation of captioning systems and unlike traditional captioning metrics encouraged diversity in generated captions. (3) We adopted SR fine-tuning and showed that the rich MLE initialization provided by VCB is important. We uncovered a trade-off between caption faithfulness and retrieval performance plaguing current SR approaches and designed a training recipe to address it. Specifically, we employed a curriculum over the number of hard negatives to more optimally leverage the SR reward, and fine-tune both the visual and language components of the captioning system. We demonstrated that this training recipe and VCB circumvents the aforementioned trade-off. With the same model architecture, our approach achieved significant performance improvements over vanilla SR (Dessì et al., 2023) across a dataset with 99 random distractors, and set the state-of-the-art

on ImageCoDe (Krojer et al., 2022). On TrueMatch, our approach outperformed powerful MLLMs that are 1-2 orders of magnitude larger and trained on large datasets, while improving over vanilla SR by up to +15%.

## 7 Limitations and Future Scope

Self-retrieval (SR) is heavily reliant on a scorer to identify and retrieve the image that best matches the generated text. We employ CLIP as a scorer, which has been shown to struggle with capturing extremely fine-grained and compositional visual details. However, it is fair to expect that the reliability of SR evaluation should improve steadily over time with better vision-language models.

Additionally, as Visual Caption Boosting uses LLMs and MLLMs for generating fine-grained descriptions, it is prone to hallucinations. However, as shown, our anchoring strategy mitigates this to a large extent.

As future work, exploring further applications of the SR fine-tuning presents a promising research avenue. MLE initialization with improved captions is also a recurring theme and played an important role in our work – this too deserves further investigation while training MLLMs.

## Acknowledgments

This project was supported in part by funding from SERB SRG/2023/002544 and an Adobe Research gift. We thank Shivanshu Sharma for partial assistance with compute. We thank Lakshmipathi Balaji, Haran SK Raajesh, Naren Akash RJ, and Vansh Agarwal for assistance with the human study.

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

# A Visual Caption Boosting

## A.1 Prompting Strategy

In this section, we discuss prompting strategies used to produce BlendCap, the visual caption, and HolisticCap. We provide the system prompt along with the in-context examples we utilize. In order to generate captions that holistically describe the image, we use the following two step process:

(a) **BlendCap.** First, we prompt Mistral-7B (Jiang et al., 2023) to reduce redundancy, preserve all factual information obtained from human annotated captions, and not introduce any new information other than what is mentioned in the reference set. This ensures that the blended caption remains factual and anchored to the human annotations:

---

**System prompt :** You are given a reference set of 5 image captions. Create a holistic caption to describe the same image by incorporating relevant information from all the reference captions. Ensure that the generated caption minimizes redundancy by excluding overlapping information found in the reference set. Do not include additional information and adjectives other than what is given in reference set. Stop after the first period and do not ask any additional questions. Just complete the text. Three examples are shown below :

**In context examples :**
**User :** Reference caption 1: A plate full of food with an assortment of food on it.
Reference caption 2: There is meat and vegetables on a white and brown plate.
Reference caption 3: A plate on a table that hasfood on it.
Reference caption 4: A plate with some steak, carrots, and sliced fried potatoes.
Reference caption 5: A plate of food on a table.
**Assistant :** A white and brown plate on a table with an assortment of steak, carrots, sliced fried potatoes, and vegetables.

**User :** Reference caption 1: A park bench on the side of a lake.
Reference caption 2: A lone bench sits atop a hill looking over the river.
Reference caption 3: A wooden bench sitting on top of a sandy beach.
Reference caption 4: A bench on a river bank in the countryside.
Reference caption 5: A wood bench is sitting in front of a river.
**Assistant :** A wooden bench sits atop a hill in front of a river overlooking the water and the surrounding countryside.

**User :** Reference caption 1: A bathroom with a toilet sitting next to a sink.
Reference caption 2: A white sink and toilet in a room.
Reference caption 3: A bathroom with a mirror, sinks, toilet and toilet roll.
Reference caption 4: A bathroom that has a toilet sink and mirror in it.
Reference caption 5: A bathroom with a toilet next to a sink.
**Assistant :** A bathroom with a toilet next to a white sink, mirror and toilet roll.

---

(b) **Visual Caption.** We concurrently use InstructBLIP (Dai et al., 2024) to produce dense visual descriptions for the given image. We find that the following simple prompt performs well.

---

**System prompt :** Briefly describe the content of the image.

---

(c) **HolisticCap.** We then prompt Mistral-7B to expand the blended caption with the visual description generated by the MLLM. Notably, the LLM is prompted to preserve human annotated information when there is a conflict of information between *BlendCap* and *Visual Caption*. Only 3 "few-shot" examples are shown here for brevity.

**System prompt :** In this task, your objective is to improve "Correct caption" by incorporating any additional information present in "New caption". If there are conflicting details between the two captions regarding the objects or their attributes, strictly use information from "Correct caption". "New caption" can have incorrect information sometimes but "Correct caption" will always be correct. The focus is on transferring missing information, including adjectives and nouns, from "New caption" to "Correct caption". Refrain from seeking further clarification and explanations; simply generate the caption. Three examples are shown below :

**In context examples :**
**User :** Correct caption: A man reaches up with a tennis racquet to hit an approaching ball in a tennis court.
New noisy caption: In this image, a woman is playing tennis on a purple court. She is wearing a white shirt and a blue short, and she is holding a tennis racket. The tennis ball is in the air as she prepares to hit it with her racket.
**Assistant :** A man wearing a white shirt and blue shorts reaches up with a tennis racquet to hit an approaching ball in a purple tennis court.

**User :** Correct caption: A white fire hydrant sits in front of an old couch on a sidewalk in front of a house.
New noisy caption: In this image, there is a black fire hydrant sitting on the sidewalk in front of a brick building. The fire hydrant is positioned next to a couch, which is placed on the sidewalk in front of the building.
**Assistant :** A white fire hydrant sits in front of an old couch on a sidewalk in front of a brick house.

**User :** Correct caption: A young boy poses next to a wall with writing on it, smiling and holding bags.
New noisy caption: In this image, a young girl is sitting in front of a graffiti-covered wall, wearing a red shirt. She is holding a box of crayons and smiling at the camera.
**Assistant :** A young boy wearing a red shirt poses next to a graffiti-covered wall with writing on it. He is holding bags of crayons and smiling at the camera.

## A.2 Human Study: Quantifying Hallucinations in VCB

We conduct a human study to investigate the extent of hallucinations in BlendCap and HolisticCap. We randomly sample 60 images from MSCOCO and for each image-caption pair, a human is asked to count the number of hallucinations present in the caption. Three types of hallucinations are considered: (1) objects, (2) attributes, and (3) egregious statements, *i.e.*, a glaringly incorrect statement that is unambiguously false.

**BlendCap** contains only 4 object and 3 attribute hallucinations. Out of the 7 hallucinations, 5 are due to incorrect COCO human annotations and only 2 (3.3%) are induced by the LLM. This demonstrates the reliability of BlendCap in combining human annotations into a denser caption.

**HolisticCap** contains only 9 instances of hallucinations (4 object and 5 attribute) across 7 captions, compared to VisualCap's 16 instances (across 12 captions). Additionally, VisualCap contained 3 egregious statements that were removed in HolisticCap. Furthermore, 7/60 captions in HolisticCap contain hallucinations, reflecting a 41.6% reduction compared to VisualCap's 12/60 captions. This demonstrates VCB's effectiveness in anchoring the MLLM descriptions (Visual Captions) into human annotations (BlendCap) to create HolisticCap.

## A.3 Token Statistics for Captions

We present some caption statistics for Visual Caption Boosting compared to original COCO captions in Table 9. While Visual Captions are longer than HolisticCap, they may contain erroneous details and

verbose language modelling priors as shown in Figure 2. Both BlendCap and HolisticCap are generally more descriptive than COCO, containing more words. Figure 4 shows the histogram of the number of tokens, tokenized using the CLIP tokenizer, for COCO, BlendCap and HolisticCap.

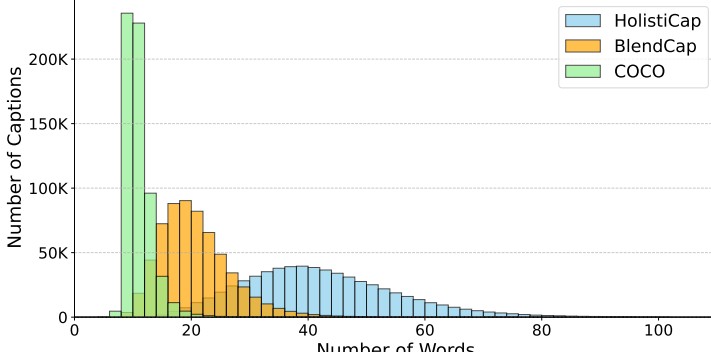

Figure 4: Histogram of number of words for COCO, BlendCap, and HolisticCap. HolisticCap is more descriptive with 41.5 words per caption on average, while COCO only has 10.5 words on average.

Table 9: Statistics of different captions. The mean and standard deviation of the number of words and tokens (post CLIP tokenization) are reported. Note the 4× difference between the original COCO and HolisticCap. Also note that the verbosity of Visual Captions is reduced (by about 10) after combining with BlendCap to create HolisticCap.

| | Dataset | Words/Cap | Toks/Cap |
|---|---|---|---|
| 1 | COCO | 10.5 ±2.4 | 13.5 ±2.7 |
| 2 | BlendCap | 20.1 ±6.0 | 25.4 ±7.0 |
| 3 | HolisticCap | 41.5 ±13.1 | 50.2 ±15.4 |
| 4 | Visual Caption | 51.1 ±14.4 | 59.2 ±12.8 |

# B   Bag Creation, Curation, and Curriculum Details

## B.1   Bag Creation and Curation

Algorithm 1 presents the process for creating bags of highly similar images that is used for both TrueMatch (Section 3.1) and SR fine-tuning (Section 4.6).

We take the bags created by Algorithm 1 on the validation and test sets of COCO (10,000 images) and use Algorithm 2 to curate bags of highly similar images for TrueMatch. The curated bags only consider the bag with the highest intra-bag similarity for a given visual concept.

## B.2   Training Bags used for Self-Retrieval

We utilize our bag creation algorithm (Algorithm 1) to create training bags. However, for each row in the sorted cosine similarity matrix $D$ we only consider the top 200 retrievals, $i.e.$ `D[:, :200]`. Each training bag of size $s$ is comprised of a given query image and the top $s-1$ images it retrieves from $D$ such that none of the retrieved images have been added to any other bag before. This ensures that all the bags are unique and each image is seen exactly once in an epoch.

Unlike TrueMatch where we select the hardest bag corresponding to specific visual concepts in the CLIP embedding space, we relax the constraint of only sampling the sub-cluster (bag of images) with the highest intra-cluster similarity.

---

**Algorithm 1** Candidate Bag Creation

---

1: **Input:**

      Output bag size $s$.

      Set of $N$ images, each with $M$ captions: $\mathcal{I} = \{(i_1, c_1^1, \ldots, c_1^M), \ldots, (i_N, c_N^1, \ldots, c_N^M)\}$.

2: **Output:**

      List of $T$ bags $\mathcal{B} = [b_1, b_2, \ldots, b_T]$ where each bag $b \subset \mathcal{I}$.

      $\mathbf{A} = [\alpha_1, \alpha_2, \ldots, \alpha_T]$ are the intra-bag similarities.

3: **Multimodal feature extraction:**

4: **for** each image $(i, c^1, \ldots, c^M)$ **do**

5:     Compute image embedding $\mathbf{z} = \texttt{CLIP.encode\_image}(i)$.

6:     Compute text embedding $\mathbf{t} = \frac{1}{M}\sum_{j=1}^{M} \texttt{CLIP.encode\_text}(c^j)$.

7:     Compute multimodal embedding $\mathbf{m} = \texttt{concat}(\mathbf{z}, \mathbf{t})$, $\mathbf{m} \in \mathbb{R}^d$.

8: **end for**

9: Compute the cosine similarity matrix $D \in \mathbb{R}^{N \times N}$ based on multimodal embeddings $[\mathbf{m}_1, \ldots, \mathbf{m}_N]$.

10: **Bag Creation:**

11: $\mathcal{B} \leftarrow [], \mathbf{A} \leftarrow []$

12: **for** each image $i_r$, $r = [1, \ldots, N]$ **do**

13:     Create a bag $b_r$ with $i_r$ and corresponding top-scoring $s{-}1$ images from row $D_r$.

14:     Compute intra-bag similarity between $i_r$ and $s{-}1$ images in the bag: $\alpha_r = \text{mean}_k(\cos(\mathbf{m}_r, \mathbf{m}_k))$.

15:     Update $\mathcal{B} \leftarrow [\mathcal{B}, b_r]$ and $\mathbf{A} \leftarrow [\mathbf{A}, \alpha_r]$.

16: **end for**

---

**Algorithm 2** TrueMatch: Automated Bag Curation

---

1: **Input:** List of $T$ bags $\mathcal{B} = [b_1, \ldots, b_T]$ and corresponding intra-bag similarities $\mathbf{A} = [\alpha_1, \ldots, \alpha_T]$

2: **Output:** A curated list of bags $\mathcal{Q}$ with highly similar images used in TrueMatch.

3: **Benchmark Creation:**

4: Sort all bags $\mathcal{B}$ in descending intra-bag similarity $\mathbf{A}$.

5: Initialize the set of visited images $\mathcal{V} = \{\}$ and curated list of bags $\mathcal{Q} = []$

6: **for** each bag $b \in \mathcal{B}$ **do**

7:     **if** $b \cap \mathcal{V} = \varnothing$ **then**

8:         Update $\mathcal{V} \leftarrow \mathcal{V} \cup b$

9:         Update $\mathcal{Q} \leftarrow [\mathcal{Q}, b]$

10:     **end if**

11: **end for**

---

Table 10: Self-retrieval fine-tuning of GPT-2 and CLIP (SR-LV) for ClipCap that is MLE initialized with HolisticCap. Values indicate R@1. (Green) numbers indicate absolute improvement and (red), a degradation.

|   | Method | CIDEr | TrueMatch RD100 | #3 | #5 | #7 |
|---|--------|-------|------|-----|-----|-----|
| 1 | Rand | 26.5 | 90.2 | 62.6 | 52.7 | 51.2 |
| 2 | Bag2 | 26.0 (0.5) | 90.5 | 64.8 | 54.0 | 55.1 |
| 3 | Bag3 | 26.0 (0.5) | 90.7 | 64.8 | 55.4 | 55.3 |
| 4 | Bag5 | 25.2 (1.3) | 90.9 | 65.0 | 55.8 | 52.8 |
| 5 | Bag7 | 25.7 (0.8) | 90.8 | 65.1 | 56.7 | 55.3 |
| 6 | Bag10 | 25.3 (1.2) | 90.9 | 65.1 | 55.8 | 54.1 |
| 7 | Bag15 | 24.9 (1.6) | 91.0 | 66.5 | 59.2 | 56.5 |
| 8 | Bag20 | 24.8 (1.7) | 91.1 | 66.5 | 58.8 | 57.3 |
| 9 | BagCurri | 29.7 (3.2) | 91.3 | 65.2 | 57.1 | 57.1 |

### B.3 BagCurri: Designing a Self-Retrieval Learning Curriculum with Bags

We design a curriculum that varies the bag sizes during training, gradually increasing the bag size with each epoch as shown in Figure 5. Since the reward is the cross-entropy of matching the generated caption with the target image Dessì et al. (2023), mining multiple hard negatives yields a flatter softmax distribution leading to stronger learning signal. Hence, increasing the bag size introduces makes the task of SR more challenging. However, it is important to increase bag sizes gradually to ensure that the task is not too hard, and prevent a collapse of caption faithfulness (Section 4.6).

To assess the role our curriculum plays in preventing this collapse, we conduct an ablation study. Specifically, we compare our curriculum against training with bags of a fixed size for the entire SR fine-tuning process. Table 10 demonstrates that our carefully designed curriculum is responsible for preserving caption faithfulness. While training with hard negatives without a curriculum improves retrieval performance on TrueMatch (rows 2-8), it fails to maintain the CIDEr score. In contrast, BagCurri not only yields substantial gains over training with random distractors (row 1) but also enhances CIDEr by +3.2%. This highlights the effectiveness of our carefully designed curriculum in preventing the collapse of NLG metrics while improving retrieval performance. Note, this is different to previous SR approaches (Dessì et al., 2023) as presented in Section 4.4.

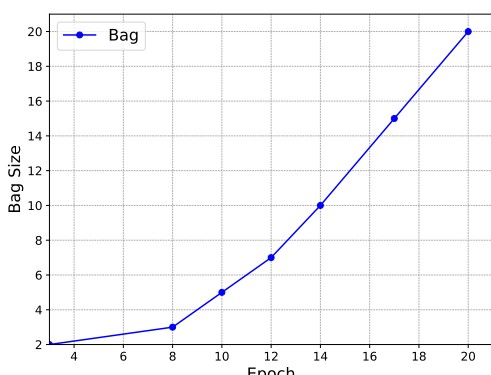

Figure 5: Our curriculum over bag sizes during training.

## C Self-Retrieval Fine-tuning

### C.1 Discussion on Optimization Strategies

Similar to (Pinto et al., 2023), we find that both optimization strategies, MLE and REINFORCE, discussed in Section 4.1 have complementary strengths and weaknesses for the task of image captioning.

**MLE.** Training the captioner on MLE results in powerful models capable of capturing complex data distributions. The task of next-token prediction with current language models is both efficient (due to teacher forcing) and scalable (Kaplan et al., 2020). However, it creates captioning systems that prefer generic captions (Wang et al., 2020), reusing a small vocabulary to describe different images with fine-grained visual differences (see Figure 1 and Section 4.3).

**Reinforce.** While fine-tuning with SR alleviates these issues (Section 4), it comes with its own set of challenges. Not only are the tokens sampled sequentially but optimizing REINFORCE is unstable, necessitating

smaller learning rates. This results in an extremely slow training process. Notably, using REINFORCE from scratch is infeasbile due to the vast action space of sampling a token and reward sparsity. However, using a pretrained MLE model provides a good initial sampling strategy and requires relatively few optimization steps to perform well on SR. This is consistent with previous works such as Rennie et al. (2017).

In Reinforcement Learning jargon, the gradients computed with REINFORCE are used directly to optimize the policy network (captioning system) in prioritizing actions (next-token prediction) that minimize the negative expected reward $\mathbb{E}_{c \sim P_\theta(\cdot|i)}[-R(c, i, \mathcal{D})]$.

## C.2 REINFORCE baseline

The variance of the gradient estimate in REINFORCE is commonly reduced by subtracting a baseline $b$ from the reward $R$. This stabilizes training by reducing the variance of the estimated gradient. We adopt a running mean of past rewards as our baseline similar to Dessì et al. (2023) as they find that it outperforms computing the reward with greedy decoding as a baseline. The latter requires sampling two outputs for one training input, one to estimate the gradient and other to compute the baseline. However, when the reward includes CIDEr, we adopt a greedy baseline.

## C.3 Overview of Hyperparameters

We provide a brief overview of hyperparameters for different optimization stages: MLE (Table 11) and REINFORCE (Table 12). Notably, CIDEr optimization for HolisticCap is done with learning rate $10^{-7}$.

Table 12: REINFORCE optimization settings.

| Base params | |
|---|---|
| Batch size: | 100 |
| Schedule: | constant |
| Total steps: | 23 000 |
| Warmup steps: | 0 |
| *SR reward* | |
| Learning-rate: | $9 \cdot 10^{-8}$ |
| *CIDEr reward* | |
| Learning-rate: | $1 \cdot 10^{-6}$ |
| *CIDEr + SR reward* | |
| Learning-rate: | $1 \cdot 10^{-7}$ |

Table 11: MLE pretraining settings.

| *MLE Pretraining* | |
|---|---|
| Batch size: | 40 |
| Schedule: | Linear Decay |
| Learning-rate: | $2 \cdot 10^{-5}$ |
| Total steps: | 30 000 |
| Warmup steps: | 1 000 |

## C.4 Fine-tuning CLIP hurts Caption Faithfulness

Table 4 shows that SR-V is substantially more discriminative compared to SR-L. However unlike SR-L, SR-V fails to preserve caption faithfulness through early stopping. To verify that the observed deterioration isn't due to the tendency of NLG metrics to penalize discriminability (Wang et al., 2020), we train SR-L until it achieves the same RD100 score as SR-V for each dataset. This ensures a fair comparison of both methods' abilities to preserve CIDEr, as presented in Table 13. We clearly see that SR-L (col 2) attains a significantly higher CIDEr score compared to SR-V (col 3) for all caption types, demonstrating that the deterioration in NLG metrics observed in Table 4 is directly caused by CLIP fine-tuning.

Table 13: Extended SR-L fine-tuning until its R@1 on RD100 becomes equal to SR-V fine-tuning R@1 when trained for 20 epochs.

| Data | MLE | SR-L | SR-V |
|---|---|---|---|
| COCO | 107.8 | 104 | 97.2 |
| BlendCap | 80.2 | 77.3 | 72.4 |
| HolisticCap | 26.6 | 26.4 | 25.6 |

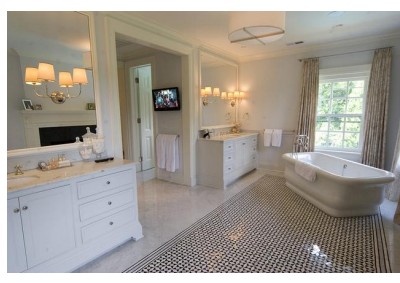
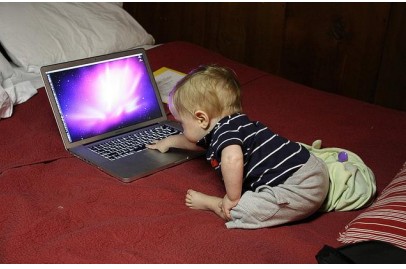
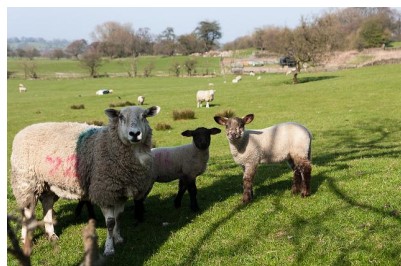

383443

**Caption** : A spacious and well-appointed bathroom features two sinks, a bathtub, and a mirror.

**CIDEr** : 0

470005

**Caption** : A baby wearing pajamas sits on a bed while using a laptop computer.

**CIDEr** : 4

410428

**Caption** : Three sheep, two of which are adults and one a baby, stand together in a grassy field.

**CIDEr** : 0

Figure 6: **Failure of CIDEr**. COCOIDs in the test set where CIDEr fails to correctly evaluate captions against HolisticCap reference set.

## C.5 Scaling CIDEr during SR optimization

The weighting of the rewards when jointly optimizing CIDEr and SR (Luo et al., 2018) determines the trade-off between faithfulness and discriminative ability of the captioning system. We MLE pretrain the captioner with HolisticCap and fine-tune both the LLM and CLIP components (SR-LV) with BagCurri. Table 14 presents an ablation over different values of $\lambda$ while scaling CIDEr in the joint reward $R = \text{SR} + \lambda \cdot \text{CIDEr}$. As expected, increasing $\lambda$ *i.e.* heavily weighing CIDEr leads to improved caption faithfulness. Interestingly, using a smaller $\lambda$ however doesn't always improve TrueMatch scores, as $\lambda = 0.7$ (row 5) outperforms $\lambda = 0.1$ (row 2). We adopt $\lambda = 0.5$ for our experiments (Section 4.8), as it results in the best performing model achieving an optimal balance between retrieval performance and caption faithfulness.

Table 14: Ablation over different lambda values for HolisticCap pretraining and SR-LV with BagCurri fine-tuning. We choose $\lambda$=0.5 as our best model.

|   | $\lambda$ | NLG | RD100 | TrueMatch | | |
|---|---|---|---|---|---|---|
|   |   | CIDEr | R@1 | #3 | #5 | #7 |
| 1 | 0 | 29.7 | 91.3 | 65.2 | 57.1 | 57.1 |
| 2 | 0.1 | 29.8 | 92.4 | 64.8 | 54.0 | 55.1 |
| 3 | 0.3 | 33.3 | 92.0 | 68.2 | 61.2 | 57.1 |
| 4 | **0.5** | 38.0 | 92.6 | 67.7 | 58.7 | 57.9 |
| 5 | 0.7 | 41.4 | 92.7 | 68.1 | 58.3 | 56.8 |
| 6 | 1 | 45.4 | 92.5 | 66.5 | 56.2 | 56.2 |

## C.6 CIDEr Struggles with Longer Descriptions

Notably, we observe that the scale of CIDEr scores for MLE pretrained models vary significantly across COCO, BlendCap, and HolisticCap (rows 1-3, Table 3). Consistent with Santos et al. (2021), we find that CIDEr rewards brevity, while reference sets containing longer captions have substantially lower scores. While the inefficacy of MLE in modeling longer descriptions does contribute to lower CIDEr scores, we also find that CIDEr fails to effectively evaluate MLE pretraining with HolisticCap, even assigning scores of 0 to correct captions, as illustrated in Figure 6.

## C.7 CHAIR: Quantifying Object Hallucinations during Extended SR-L Fine-tuning

We generate captions for all the 10,000 images of MSCOCO test and validation set and compute object level hallucinations using CHAIR (Rohrbach et al., 2018). CHAIR compares model-generated captions with human annotations and provides 2 scores $\text{CHAIR}_I$ and $\text{CHAIR}_S$ to quantify object and sentence level hallucinations respectively:

$$\text{CHAIR}_S = \frac{|\{\text{captions with hallucinated objects}\}|}{|\{\text{all captions}\}|}, \quad \text{CHAIR}_I = \frac{|\{\text{hallucinated objects}\}|}{|\{\text{all objects mentioned}\}|}. \quad (2)$$

Table 15: **Evaluating object hallucinations**. We use CHAIR to study the extent of object hallucinations induced by MLE training and SR-L fine-tuning.

| Dataset | Method | CHAIR$_I$ | CHAIR$_S$ | Hallucinated Objects | CIDEr |
|---------|--------|-----------|-----------|----------------------|-------|
| COCO | MLE | 11.66 | 8.38 | 1255 | 107.8 |
| | SR-L | 9.79 | 7.03 | 1033 | 108.2 |
| | SR-L 100 epochs | 9.63 | 7.33 | 1102 | 92.4 |

**MLE training** encourages the captioner to generate generic phrases for different images that may not be aligned with the image content, resulting in increased object hallucinations.

**SR-L fine-tuning** may struggle with attribute binding, but it generates captions that are more discriminant and specific to the image, resulting in a significant reduction in object hallucinations.

**Extended fine-tuning with SR-L** for 100 epochs in addition to reducing caption faithfulness, also increases both the absolute count and percentage of hallucinated objects.

### C.8 Investigating Caption Unfaithfulness in SR Fine-tuning

To improve our understanding of the trade-off presented in Section 4.4 during SR fine-tuning, we further investigate the deterioration of the NLG metrics. Dessì et al. (2023) attribute this deterioration to the model diverging from the GT distribution during SR-L fine-tuning. They argue it to be desirable property as it allows the captioner to generalize to a wider range of captioning distributions. We investigate this claim by qualitatively analyzing image-caption pairs (COCO test set) where SR-L exhibits poor CIDEr scores compared to MLE. We find several image-caption pairs (some are shown in Figure 7) corroborating this claim, highlighting the inadequacies of reference-based metrics. For instance, a more discriminative caption "A woman serving a hot dog to a man" scores lower on CIDEr than a more generic one, "A couple of people standing in front of a food truck". However, our analysis also reveals that models fine-tuned on SR, in an attempt to enhance retrieval, have a propensity to hallucinate which results in poor CIDEr scores (see Figure 8). This differs from the findings of Section 4.3, since the model is incorporating non-factual information to the generated captions instead of semantic details. We observe this tendency under the same experimental setting as (Dessì et al., 2023), noting a modest dip in NLG metrics due to early stopping. However, we find this effect exacerbates with longer training durations, as illustrated by the dramatic drop in CIDEr scores in Figure 3.

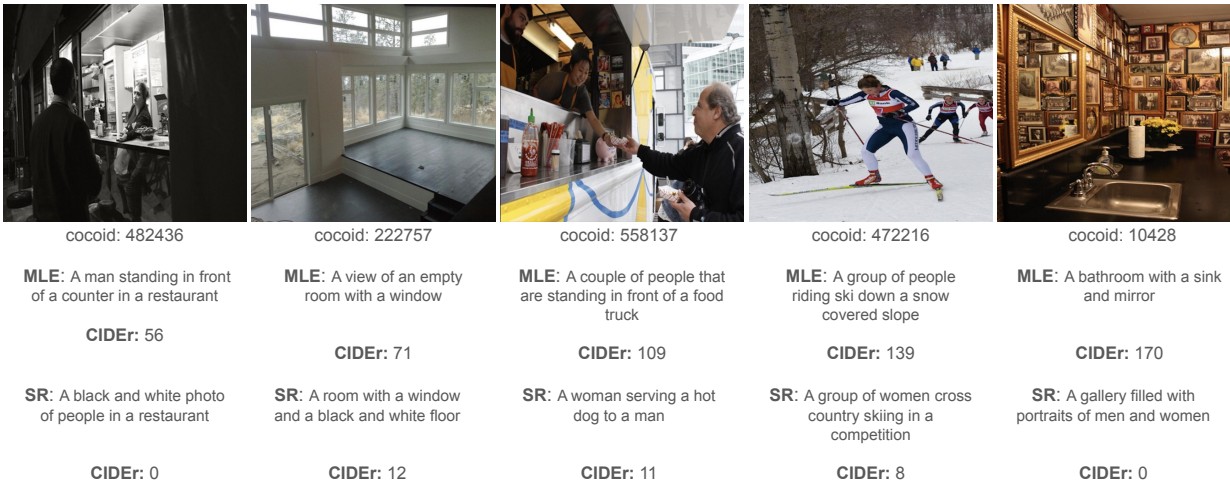

Figure 7: **Low CIDEr due to generalization**. COCOIDs in test set, where COCO SR fine-tuned caption scores poorly on CIDEr compared to COCO MLE trained model due to generated captions being different from COCO's ground truth distribution.

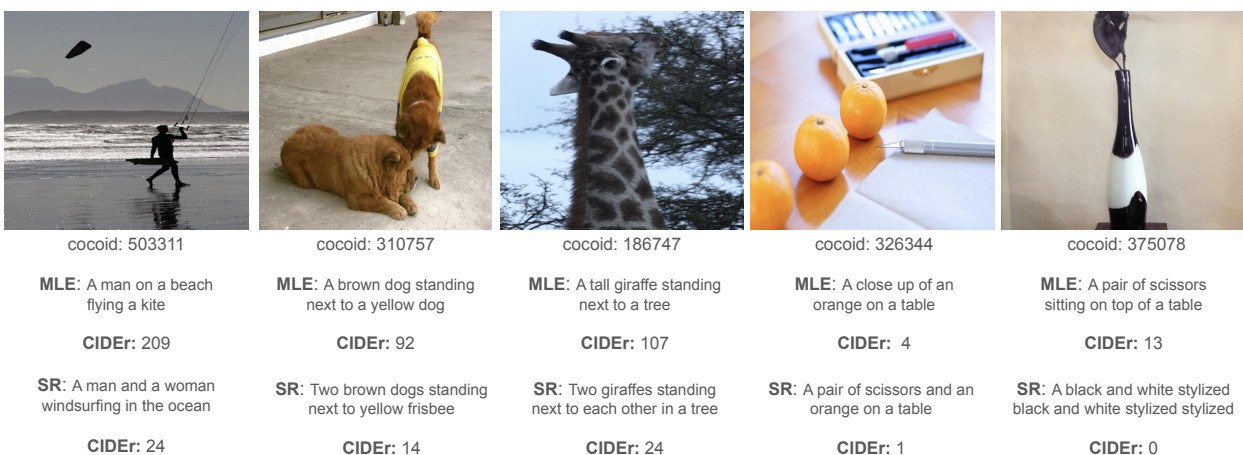

Figure 8: **Unfaithfulness through hallucination.** Some COCOIDs from the test set where SR fine-tuned captioner scores poorly on CIDEr compared to COCO MLE trained model due to nonfactual information.

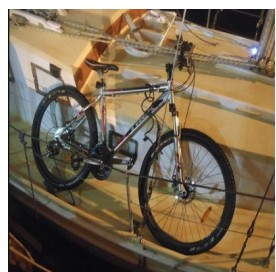 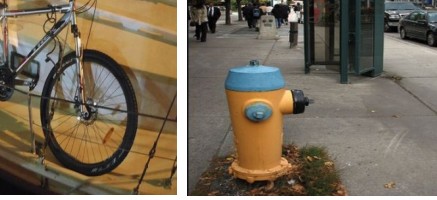 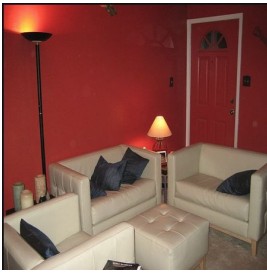 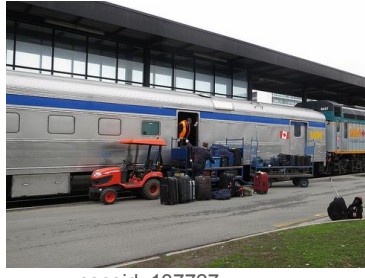

| cocoid: 536589 | cocoid: 72397 | cocoid: 560756 | cocoid: 137727 |

**SR-L**: A bicycle is parked next to a boat on a dock.

**CIDEr:** 77

**SR-V**: Two bicycles that are next to each other.

**CIDEr:** 0

**SR-V curri**: A bicycle and a car are displayed in a display case.

**CIDEr:** 12

**SR-L**: A blue fire hydrant sitting on the side of a street.

**CIDEr:** 275

**SR-V**: A blue fire hydrant sitting on the side of a street.

**CIDEr:** 275

**SR-V curri**: A blue and blue blue blue and yellow blue and yellow blue and yellow blue and yellow blue…

**CIDEr:** 0

**SR-L**: A living room filled with furniture and a couch.

**CIDEr:** 83

**SR-V**: A living room filled with red chairs and orange walls.

**CIDEr:** 13

**SR-V curri**: Three orange chairs sitting next to each other in a room.

**CIDEr:** 0

**SR-L**: A blue and grey passenger train is parked on the tracks at a train station, with a crane nearby.
**CIDEr:** 22

**SR-V**: A blue and orange train sits on the tracks at a train station, ready for passengers to board
**CIDEr:** 7

**SR-V curri**: Two blue and orange train cars are parked next to each other on the train tracks.

**CIDEr:** 0

Figure 9: SR-V fine-tuned captioner struggles with attribute binding and hallucination. Training with BagCurri (SR-V curri) worsens this tendency.

Verified Bags in TrueMatch            Bags Discarded with Manual Verification

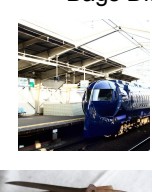 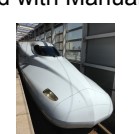 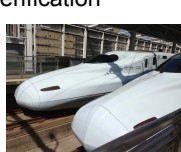

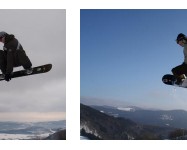 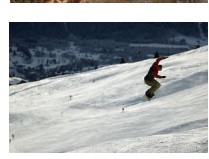 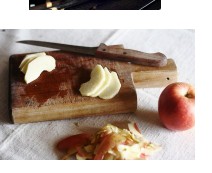 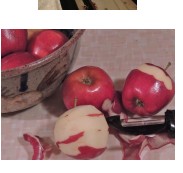 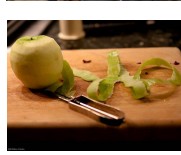

Figure 10: **Manual filtering of bags in TrueMatch**. We manually remove bags that do not capture some aspect of fine-grained visual discrimination, even if they represent the same visual concept.

