# OpenReview forum: "No Detail Left Behind: Revisiting Self-Retrieval for Fine-Grained Image Captioning"
_TMLR — Accepted by TMLR_

### Review · Reviewer_tNaH · 2024-09-28

**Summary Of Contributions:**

There are three main contributions proposed in this work.  The first and second are a proposal to implement a curriculum for training image captioning models via self-retrieval in a way that targets the claimed observation that such methods are prone to hallucinations, with human-anchored captions, via a framework the authors call Visual Caption Boosting. The authors also propose a new benchmark which contains very similar pictures and forces models to be able to discern small details to match images to descriptions.

**Audience:**

Yes

**Broader Impact Concerns:**

The authors do well to consider potential broader impact concerns and I was impressed by how this was presented in the submission and how cultural issues are taken into consideration (in terms of what it means for different cultures to notice and report in annotations of visual images).

**Claims And Evidence:**

Yes

**Requested Changes:**

My requested changes are non-critical and more in aid of increasing the submissions readability. I would be happy to see the submission accepted as is, if there were not time/interest available in making the changes.

Figure 2 gets referred to prior to the definitions of ‘HolisticCap’ and ‘BlendCap’ but these are used in the figure caption. This happens a bit too early and is a bit confusing. I would recommend either amending the caption to define them or use those terms in the figure somewhere. It’s not an impediment to understanding in any strong sense, but it’s not clear when reading whether the specific term there directly refers exactly to what’s implied in the image or could be a derived system that is yet to be explained. I think the easiest solutions is to say ‘holistic / blended captions’ (not ‘HolisticCap’ / ‘BlendCap’) in the figure description, for clarity, and then define those other terms later on the same page.

**Strengths And Weaknesses:**

This is a fantastically written paper. It has virtually impeccable English and it was a pleasure to read something that has obviously been curated and prepared with careful attention to detail.

I have some thoughts I jotted down during reading, based on a selected figure or two (mainly Figure 1 as it sets up the conceptual reason for the proposed work, while all follow-up work is carefully addressing issues raised from this point).

Figure 1 is intended to give a concrete example to illustrate the current “problem” and how their approach solves it. The COCO MLE versions give the same caption to the same image (as they are all visually similar). The COCO SR parrots the COCO MLE version, but gets the 2nd picture wrong (there are not two people in this image). The proposed method (3rd set of captions) gets more details correct by mentioning the presence of gloves and the colour of the bike. However, it feels like the captions are divergent for the sake of text diversity alone. Why in some cases but not others could you call them motorcyclists? Intuitively, I feel like if I wanted a system to caption some images, the visual similarity of the images would make me prefer a system that was broadly consistent. If I wanted to use a machine learning system based on such captions, I’d be a bit concerned there was a tendency to mention specific items in images for some captions but not others (the presence of clothing description in one caption, but not the others). This means for the sake of being ‘identifiable’ (describing an image in one way that isn’t applied to the others) it could result in a patchwork-like situation where different details are highlighted in different images to pass the task.

I think whether this is desirable or not depends on your use ase. If my system should learn many details across many images, the extra details in some images probably is desirable and useful if this approach was used to create training data. However, on an individual case-by-case basis if this was part of a final user-facing system.

This sort of sets me up to wonder just how much of a problem this actually is, with this being the flagship example the authors have chosen to promote. It is nice to see that the only caption for all variants that doesn’t automatically assume the presence of a man is a caption proposed in the authors current work (3rd caption). I just don’t think the authors are being fair when they imply the MLE captions don’t meaningfully capture the semantics of each image.

---

In Figure 2, how do you derive what is a hallucination? Specifically, the example includes underlined text (red line) and the system should default to human annotations in these cases and not include them in the holistic caption. I wasn’t sure how the system did this and would like to know. Is this derived when the MLLM visual description includes details not present in ANY of the human annotator captions or is there some sort of different system?

----

You can always design a benchmark that will emphasise particular properties of a proposed system, so evaluation on TrueMatch over the larger MLLMs is not painting the full picture, unless that benchmark encompasses most things that are desirable in a captioning system. I didn't get the sense that is what TrueMatch claimed to be, so I think the authors could couch some descriptions a bit better with this in mind. I got the sense the reported better metrics on their own design of a benchmark was perhaps implying more of an external validation than is perhaps warranted. However, this derives from my earlier questions around Figure 1. All in all, this is quite a minor point.

----

In conclusion, this is an exceptionally strong submission and it was a pleasure to read. My comments above relating to potential weaknesses are relatively minor. Overall, it's a very impressive piece of work and I am happy to have had the opportunity to review it.

---

> ### Author Response · Authors · 2024-10-05
> **Thank you, a little delayed in responding**
>
> Dear reviewer tNaH, thank you so much for your tremendous support and review of our work. We really appreciate it. Our lead author is sick and had to visit the hospital over the last week. While we have an initial response ready, we request for some more time to respond to your valuable suggestions. Thank you for understanding!

---

> > ### Comment · Action_Editor_KGor · 2024-10-06
> >
> > Dear authors,
> >
> > We are still waiting for two more reviews (I sent a reminder a few days ago, but they are still irresponsible).
> >
> > The official author-reviewer discussion period will start after we receive all three reviews; hence, you don't need to hurry to respond.
> >
> > Also, if you need more time after the official discussion period starts, please don't hesitate to contact me.
> >
> > AE

---

> ### Author Response · Authors · 2024-11-10
> **Response for tNaH (R1)**
>
> Thank you so much for your kind words of appreciation. We were truly happy and excited to hear that you enjoyed reading our paper. We also thank you for the feedback on improving the paper’s readability by changing "HolisticCap/BlendCap" to "Holistic/Blended captions" in the Figure 2 caption. We have made that change. Below, we respond to your thought provoking questions about the paper.
>
> **1. Does SR lead to divergent captions only for the sake of text diversity? Is consistency desirable?**
>
> We expect captioners to describe fine-grained details that are likely to be discriminative. As our model does not see the other images (distractors) of a bag, it cannot guarantee a priori that the generated caption is able to uniquely retrieve the image. Not seeing other images is desirable as it prevents collapse to silly artifacts (e.g., we do not want a caption to state that the bottom part of the middle image is darker than the left image, in Figure 1). Similarly, paraphrasing the caption for text diversity is also unlikely to improve performance.
>
> In Figure 1, both COCO MLE and COCO SR do not generate discriminative captions resulting in 0% SR accuracy. In contrast, our method achieves 66% accuracy by capturing discriminant visual details, such as "jumping high in the air" for image 2 (middle) and "red and white dirt bike" for image 3 (right). Notably, our method fails to uniquely describe image 1 (left), showing that text diversity (e.g., "riding on a cloudy day") and non-discriminant but correct details (e.g., "wearing gloves") are insufficient for solving SR. We have updated the Figure 1 caption to clarify this.
>
> As SR aims to instill fine-grained information, some diversity in captioning is expected due to stochasticity of sampling. We believe that our model "knows" many visual details, verified by generating 10 captions using beam search, but tends to only describe a few when generating a specific caption. We think this is desirable as it produces succinct captions that are also likely to be discriminative.
>
> **2. Does MLE capture image semantics?**
>
> We agree with the reviewer that MLE captures some image semantics. However, MLE trained captioners encourage generation of statistically probable phrases leading to generic captions, especially when trained on datasets such as MSCOCO. We have updated the Figure 1 caption slightly to indicate that the findings are specific to this example.
>
> **3. Questions about Figure 2**
>
> _What is a hallucination?_ Since BlendCap comprises reliable human annotations, we consider any visual information in the Visual Caption that _conflicts_ with BlendCap as a hallucination. The colored annotations and red lines are added by us (not the VCB system) to help the reader. We make this explicit in the caption.
>
> _How does the system know what to include/exclude in HolisticCap?_ During the anchoring process, we prompt the LLM and provide 6 few-shot examples (3 shown in Appendix A.1 for brevity) to remove visual details that conflict with BlendCap and non-semantic verbose language. As discussed in response 1 to reviewer Zz97, this leads to a 41.7% reduction in hallucinations of the VisualCap.
>
> **4. What aspects of a captioning system does TrueMatch evaluate?**
>
> A good captioning system should be: (i) grammatically correct; (ii) descriptive (describe visual elements in the image); and (iii) discriminative (describe fine-grained visual details). Current metrics evaluate grammatical correctness (BLEU score, CIDEr) and descriptiveness (CLIP-Score, CIDEr, SPICE) but fall short in assessing discriminativeness. We propose SR evaluation with TrueMatch to measure how discriminative and therefore fine-grained the captions are. We believe that a comprehensive evaluation should include SR along with another metric that captures descriptiveness and grammar (e.g., CIDEr).

---

> > ### Comment · Reviewer_tNaH · 2024-11-14
> > **Questions answered**
> >
> > Thank you to the authors for this well-considered response to the questions I raised. I'm satisfied that the changes implemented render the paper more understandable and clearer from the perspective of an interested reader.

---

### Review · Reviewer_Wasn · 2024-10-26

**Summary Of Contributions:**

The manuscript proposes 3 changes the pipeline for training a captioning model, towards the goal of obtaining fine-grained detailed captions that are also "correct".

1. The training data is augmented using LLMs and MLLMs to combine multiple human captions per image into one single detailed caption. This makes the training data itself capture the sort of fine-grained details one desires from captioning models. The end result is the HolisticCap variation of COCO captioning data.

2. A new evaluation metric called TrueMatch that builds on self-retrieval based evaluation. The contribution here is to mine similar bags of examples of different sizes so that the self-retrieval task is challenging from a fine-grained perspective.

3. The training pipeline is improved in many ways:
- Phase 1 trains on HolisticCap data to get a strong initial model.
- Phase 2 fine-tunes using the self-retrieval objective on challenging bags. Here the contribution is to fine-tune all model components and use a curriculum of self-retrieval bags.

The paper analyzes weaknesses in the existing approach and illustrates how the several contributions work together to solve it.

**Audience:**

Yes

**Broader Impact Concerns:**

A limitations section is included but there is no broader impact statement. COCO includes the person category so some discussion is needed as captioning people with images requires more care with regards to model fairness.

**Claims And Evidence:**

Yes

**Requested Changes:**

Discussed in the previous box.

**Strengths And Weaknesses:**

Strengths
=======
- Well written paper with lots of insights and contributions that improve performance.
- The dataset and metric would be easy to use if published (as in these are simple to adopt). The training pipeline is complicated but the complexity is justified by the empirical evidence and trade-offs involved.
- The discussion balances well the several metrics that need to be tracked and provides an interpretation for the reader that is consistent throughout.

Weaknesses
=========
- Will the HolisticCap dataset and the bags used for TrueMatch and for SR finetuning be made public?
- Will any code be published?

Clarifying questions
===============
- How was manual filtering of the bags in section 3.1 done? Examples in the appendix of what was filtered and not filtered would be helpful.
- The bag curation in section 3.1 does a sort of non-maximum supression to avoid reusing images and picking the most challenging bags. The last line additionally claims that the bags capture different aspects of visual discrimination beyond object, positioning and negation. How is this achieved?
- In section 3.3, 'we observe that HolisticCap, which uses InstructBLIP': How does HolisticCap use InstructBLIP? I didn't understand this part.
- Section 4.2 says that LoRA tuning is done, but section 4.3 starts with finetuning the langauge model. When the manuscript says SR-X is X being LoRA tuned?
- In Table 1, is "Ours (best)" HolisticCap + SR-LV + Bag-Curri + CIDEr optimization?

---

> ### Author Response · Authors · 2024-11-10
> **Response for Wasn (R2)**
>
> Thank you for your kind words of appreciation for our work. We provide clarifications below.
>
> **1. Will the code, dataset, and benchmark be open-sourced?**
>
> Yes, we plan to document and release everything. All the captioning datasets (Blended, Visual, and Holistic captions), the TrueMatch benchmark, and the training and evaluation code for various models will be released. We will also share some model checkpoints and a Gradio demo for users to play with our models.
>
> **2. How is manual filtering done?**
>
> _Bag curation as NMS_. The reviewer is correct. The bag curation process can be thought of as NMS in the embedding space. In typical NMS, overlapping bounding box detections are removed, keeping the one with highest confidence. Similarly, for each cluster in the embedding space that represents a specific visual concept (e.g., cows grazing), we select the bag with the highest intra-bag similarity (most similar images).
>
> _Manual filtering_ of bags involves verifying that the automated bag creation process creates bags with highly similar images and fine-grained differences. We discard bags with images that are deemed easy to distinguish, e.g., having an easy difference in the object or background. A couple examples are presented in Figure 10 in the Appendix.
>
> Together, the process results in bags containing highly similar images that focus on diverse concepts and capture different visual aspects such as object state or attribute, orientation, action, count, etc.
>
> **3. How does HolisticCap use InstructBLIP?**
>
> The Visual Caption is generated using InstructBLIP (see Figure 2). Subsequently, HolisticCap is generated by anchoring semantic visual details from the Visual Caption into BlendCap. We have revised Section 3.3 to improve clarity on this matter.
>
> **4. LoRA fine-tuning**
>
> Yes, in all SR-X experiments, whenever X is fine-tuned, we use LoRA adapters. Thanks for pointing out the confusion, we have updated Section 4.3.
>
> **5. In Table 1, is "Ours (best)" HolisticCap + SR-LV + Bag-Curri + CIDEr optimization?**
>
> Yes, that is correct.

---

> > ### Comment · Reviewer_Wasn · 2024-11-26
> >
> > Thank you for the reply. My concerns have been addressed.

---

### Review · Reviewer_Zz97 · 2024-10-28

**Summary Of Contributions:**

The paper proposes solutions to improve fine-grained image captioning, which often yields generic descriptions due to noisy training data and MLE training methods. The authors introduce Visual Caption Boosting to enhance captions with detailed descriptions and BagCurri, a training curriculum that optimally utilizes self-retrieval (SR) rewards. They also present TrueMatch, a new evaluation metric to assess a model's ability to distinguish subtle visual details. Overall, the approach enhances caption specificity and accuracy, outperforming existing methods on various benchmarks.

**Audience:**

Yes

**Broader Impact Concerns:**

Since this framework addresses fine-grained image description generation, there are concerns regarding the potential inclusion of undesirable content in the generated responses. Furthermore, the framework may inherit inherent biases from the pretrained LLMs used in its pipeline, raising important considerations about the model's fairness and reliability in real-world applications.

**Claims And Evidence:**

Yes

**Requested Changes:**

- I recommend revising the statement about self-retrieval guiding the captioner away from its language modeling priors (Finding 2). It would be more accurate to describe self-retrieval as "unlocking" or "unleashing" the potential for descriptive captioning. This emphasizes how it harnesses latent semantic information, enabling more nuanced and fine-grained captions.
Additionally, the results for maximum likelihood estimation (MLE) are not as low as implied; MLE maintains a respectable performance level. Acknowledging this balance would provide a clearer picture of the effectiveness of the different training approaches.

- As evidenced by Finding 3, the fine-tuning process with SR-L exhibits a tendency toward hallucination generation. The paper would benefit from a quantitative assessment of these hallucination phenomena using established vision-language hallucination metrics, providing a more rigorous evaluation of the framework's reliability.

**Strengths And Weaknesses:**

- Strengths
  - The paper addresses the crucial challenge of fine-grained caption generation in vision-language understanding.
  - The proposed approach demonstrates strong performance across benchmarks, supported by comprehensive ablation studies and empirical analyses.
  - The thorough evaluation protocol effectively validates the framework's robustness, while providing insightful qualitative examples of the model's capabilities.

- Weakness
  - A significant concern regarding this submission is the potential for hallucinations throughout the pipeline. The multi-stage data generation process, which incorporates both LLM and MLLM components, may introduce hallucinations despite the anchoring mechanism to blended captions. Such data-level hallucinations could compromise the data quality and consequently affect the validity of the entire experimental protocol.
  - A notable limitation lies in the evaluation methodology, which relies heavily on CLIP score-based retrieval. Since CLIP treats captions as bag-of-words, this metric may not adequately capture the semantic relationships and intended meanings in the generated captions, potentially providing an incomplete assessment of the model's true performance. This limitation is particularly problematic in the context of fine-grained vision-language understanding, where precise semantic relationships and nuanced details are crucial for evaluation.

---

> ### Author Response · Authors · 2024-11-10
> **Response for Zz97 (R3)**
>
> Thank you for your insightful review and appreciation of our work. We provide clarifications below.
>
> **1. Hallucinations induced by VCB pipeline may compromise data quality**
>
> We take significant steps to ensure high data quality by anchoring the MLLM description (Visual caption) to human annotations (Blended caption). Specifically, we use detailed prompts and in-context examples, teaching the LLM to ignore visual information that conflicts with human annotations (see Appendix A.1).
>
> However, the reviewer raises a valid point about LLMs and MLLMs being prone to hallucination, which we investigate through a human study. We randomly sample 60 images from our dataset and for each image-caption pair, request a human to identify three types of hallucinations: (i) wrong objects, (ii) wrong attributes, and (iii) egregious statements (glaring statements that are unambiguously wrong). We find that 7/60 Blended Captions contain hallucinations. Of these, 5 stem from incorrect COCO human annotations and only 2 are induced by the LLM. Furthermore, anchoring the Visual Captions in human annotations reduces the hallucinations from 12 (in VisualCap) to 7 (in HolisticCap), a reduction of 41.7%. The updated Appendix A.2 presents details.
>
> Finally, we note that the principle of VCB is a model-agnostic framework, and will only improve with advancements in foundation models. We also revise the limitations section, acknowledging that the VCB pipeline is prone to hallucinations due to the use of LLMs and MLLMs.
>
> **2. Self-Retrieval evaluation is bottlenecked by the scorer.**
>
> We acknowledge this in Section 7 on “Limitations and Future Scope”. We expect SR evaluation to become more reliable with the development of fine-grained VLMs and may even incentivize their development.
>
> **3. Requested changes for Section 4.3**
>
> We agree with the reviewer that the performance of MLE-trained models is not as low compared to vanilla SR-L in Table 3. We have revised some words in Section 4.3 to reflect this. Regarding Finding 2 (and the sentence just above it), we do write that "SR unlocks latent semantic information" and are not sure what the reviewer meant by revising it.
>
> **4. Quantifying hallucinations due to SR-L (Finding 3)**
>
> The trade-off discussed in Section 4.4 is only observed when the captioner is fine-tuned with SR-L for a larger number of epochs (e.g., 100). Normally, we fine-tune with SR-L only for 20 epochs, preserving CIDEr and increasing SR performance. We have revised Finding 3 to clarify this.
>
> Based on the reviewer’s recommendation, we investigate object-level hallucinations in MLE and SR-L captions on all 10,000 images of the COCO val and test sets. We compute object hallucinations using CHAIR (Rohrbach et al., 2018), obtaining 2 scores: CHAIRi and CHAIRs, the proportion of hallucinated objects and captions respectively (lower is better).
>
> The Table below shows that the MLE trained model tends to generate generic phrases that may not be aligned with the image content, resulting in many object hallucinations. Although SR-L may struggle with attribute binding (see Figure 9), it makes the generated captions more fine-grained, resulting in a significant reduction in object hallucinations (1255 to 1033). Extended fine-tuning of the captioner with SR-L for 100 epochs increases both the absolute count (1033 to 1102) and percentage of hallucinated objects (CHAIRi, 7.03% to 7.33%). This is aligned with the findings of Section 4.4. We include this experiment in Appendix C.7.
>
> | Method    | CHAIRs | CHAIRi | absolute | CIDEr |
> | ------------- | ------ | ------ | -------- | ----- |
> | MLE      | 11.66 | 8.38  | 1255   | 107.8 |
> | SR-L      | 9.79  | 7.03  | 1033   | 108.2 |
> | SR-L-100epochs | 9.63  | 7.33  | 1102   | 92.4 |

---

> > ### Comment · Reviewer_Zz97 · 2024-11-27
> >
> > Thanks for the response and updated experiments. All of my concerns have been resolved except for the matter regarding Finding 2. I wasn't trying to repeat what was already written, but given that the MLE-trained model's performance is not low, I made this comment because the expression 'guiding the captioner away' seems inappropriate.

---

> > > ### Author Response · Authors · 2024-11-27
> > > **Revised Finding 2**
> > >
> > > Thank you for your response. We acknowledge your concern, and will revise Finding 2 in the updated pdf to:
> > > Self-retrieval steers the language model to generate discriminative captions by uncovering latent semantic information that was initially obscured by MLE.

---

### Author Response · Authors · 2024-11-10
**Revision + Overall response to Reviewers and AE**

Dear reviewers, thank you for your comments and constructive feedback that has improved our work. We have left official comments for each reviewer and have also uploaded a revised PDF with changes highlighted in blue for convenience. We hope that these changes and responses address your comments.

Dear AE, please let us know if anything else is required. We may however need an additional week due to the approaching CVPR deadline. Thank you for your understanding and help!

---

### Decision · Action_Editor_KGor · 2024-11-29

**Recommendation:** Accept as is

**Comment:**

During the review-author discussion period, all the reviewers agreed that this paper solves an interesting problem and is well-written. I also think that the contribution of the current paper is sufficient to be published as a TMLR paper.

**Audience:**

This paper tackles an interesting problem that might be intriguing to many TMLR audiences.

**Claims And Evidence:**

This paper tackles the fine-grained image captioning problem that often generates less faithfulness and even hallucinated captions due to the noisy training data and MLE training methods.

The noisy training data is addressed by data augmentation using LLMs and MLLMs (Visual Caption Boosting,). LLMs combine multiple human captions into one single detailed caption (Blended Caption) and Visual Caption is extracted by MLLMs. Finally, Holistic Caption is generated by using Blended Caption and Visual Caption as anchors.

The training method is improved by the Self-Retrieval (SR) objective function. The proposed BagCurri, a curriculum learning framework that optimally utilizes SR rewards, boosts the fine-grained image captioning task with a meaningful gap.

Finally, this paper also proposes a new TrueMatch metric, an evaluation metric based on SR. This new metric shows that many existing captioners may not perform well in fine-grained caption generation tasks.